# How is avalanche danger described in textual descriptions in avalanche forecasts in Switzerland? Consistency between forecasters and avalanche danger

Veronika Hutter[1,2*], Frank Techel[2,3], and Ross S. Purves[3]

[1]School of Life Sciences, Technical University of Munich, Germany
[2]WSL Institute for Snow and Avalanche Research SLF, Davos, Switzerland
[3]Department of Geography, University of Zurich, Switzerland
[*]affiliation at time of writing thesis

**Correspondence:** Frank Techel (techel@slf.ch)

**Abstract.** Effective and efficient communication of expected avalanche conditions and danger to the public is of great importance, especially where the primary audience of forecasts are recreational, non-expert users. In Europe, avalanche danger is communicated using a pyramid, starting with ordinal levels of avalanche danger, and progressing through avalanche-prone locations and avalanche problems to a danger description. In many forecast products, information relating to the trigger required to release an avalanche, the frequency or number of potential triggering spots, and the expected avalanche size, are described exclusively in a textual danger description. These danger descriptions are, however, the least standardized part of avalanche forecasts. Taking the perspective of the avalanche forecaster, and focusing particularly on terms describing these three characterizing elements of avalanche danger, we investigate first which meaning forecasters assign to the text characterizing these elements, and second how these descriptions relate to the forecast danger level. We analysed almost 6000 danger descriptions in avalanche forecasts published in Switzerland, and written using a structured catalogue of phrases with a limited number of words. Words and phrases representing information describing these three elements were labeled and assigned to ordinal classes by Swiss avalanche forecasters. These classes were then related to avalanche danger. Forecasters were relatively consistent in assigning labels to words and phrases with Cohen's Kappa values ranging from 0.67 to 0.87. Avalanche danger levels were also described consistently using words and phrases, with for example avalanche size classes increasing monotonically with avalanche danger. However, especially for danger level 2-Moderate, information about key elements of avalanche danger, for instance the frequency or number of potential triggering spots, was often missing in danger descriptions. In general, the analysis of the danger descriptions showed that extreme conditions are described in more detail than intermediate values, highlighting the difficulty of communicating conditions that are neither rare nor frequent, or neither small nor large. Our results provide data-driven insights that could be used to refine the ways in which avalanche danger could be communicated. Furthermore, through the perspective of the semiotic triangle, relating a referent (the avalanche situation) through thought (the processing process) to symbols (the textual danger description), we provide an alternative starting point for future studies of avalanche forecast consistency and communication.

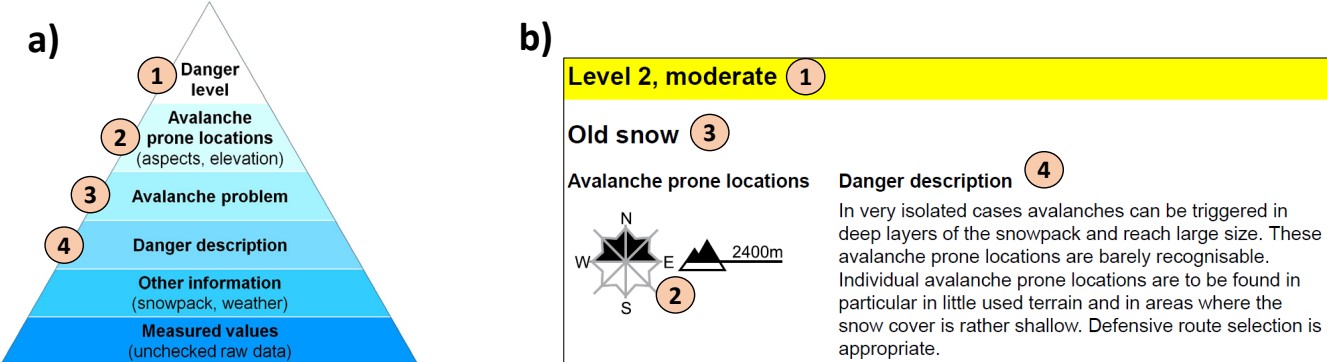

**Figure 1.** (a) In Europe, avalanche forecasts are structured according to an information pyramid (EAWS, 2017a) with the danger level at the top (1), followed by the avalanche-prone locations and the avalanche problem (2 and 3), and a danger description (4) providing further details. (b) Exemplary description of a danger region in the Swiss avalanche forecast (forecast published on 2021/02/26 at 8 am). Even though the danger description is written in present tense, it describes the expected conditions for the 24 hours following publication.

# 1 Introduction

Public avalanche forecasts, as provided in many mountainous regions, inform readers about snow and avalanche conditions at a regional scale. To effectively communicate different components of the avalanche situation, forecasts often describe expected conditions systematically, using a hierarchical information pyramid communicating the current regional danger level, the most avalanche-prone locations (*aspects* and *elevation range*, also called the *core zone*), the dominant *avalanche problem*, and using narrative text, a *danger description* (Fig. 1a and b; e.g. Engeset et al., 2018; SLF, 2020) or snowpack and weather summaries. Although the communication of danger levels, corresponding core zones and dominant avalanche problems is often standardized using common terms or symbols (e.g. in Europe: five danger levels (EAWS, 2018), five avalanche problems (EAWS, 2021b)), the degree of detail and the use of text and graphics varies considerably between forecast products issued by different (usually national) forecasting centres (Burkeljca, 2013; Engeset et al., 2018; Techel et al., 2018). Important information describing the severity of the avalanche conditions, such as likely triggers required to release an avalanche, the frequency with which such triggering spots will be found in a region, the specifics of the likely locations of these triggering spots, and the expected avalanche sizes are communicated in various ways (Tab. 1). Even though this information defines the avalanche danger level, its communication is much less standardized. In North America, this information is often provided through a combination of key words and graphics. In Europe, however, narrative textual danger descriptions are much more common (Tab. 1, cf. Fig. 1b), though in Norway a tabular format has been adopted (e.g. in Norway; Engeset et al., 2018).

Public avalanche warning services in Europe aim to provide efficient and effective forecasts to their users (e.g. EAWS, 2017b). To achieve this goal, forecasts in general must provide credible information of value to the user (e.g. Williams, 1980; Gordon and Shaykewich, 2000). Value, however, is directly influenced by forecasts being consistent and accurate (Murphy,

**Table 1.** Overview showing examples of how information regarding contributing factors of avalanche danger is presented in avalanche forecast products. For each forecasting centre, a specific, randomly selected forecast product is provided in the supplementary material. *the factors characterize the avalanche problem(s) separately. **the description of the factors refers to the main avalanche problem(s) relevant for the avalanche danger assessment, and/or avalanche danger as a whole. The description of all the factors characterizing the avalanche problem or avalanche danger within the text is not compulsory.

| Forecast in | factors | communication through | (further) text elements in forecast |
|---|---|---|---|
| Canada | likelihood, avalanche size* | graphic, key word | avalanche problem description, travel and terrain advice, forecast details |
| Colorado (United States) | likelihood, avalanche size* | graphic, key word | summary, forecast discussion |
| France | *(not separately listed)*** | key word summary | snow quality, snowpack stability |
| Norway | trigger, distribution, avalanche size, probability* | bullet points with key words | avalanche danger assessment, snowpack summary, weather |
| Bavaria (Germany), Euregio (Austria, Italy), Switzerland | *(not separately listed)*** | narrative danger description | snowpack, weather, outlook |

Canada: Avalanche Canada; Colorado: Colorado Avalanche Information Centre (CAIC); France: MétéoFrance; Norway: Norwegian Water Resources and Energy Directorate (NVE); Euregio: Province of Tyrol, Autonomous Provinces of Bolzano-South Tyrol and Trento; Switzerland: WSL Institute for Snow and Avalanche Research SLF

1993). In recent years, this fact has increasingly been recognized in avalanche forecasting. To date, most effort exploring quality and consistency of avalanche forecasts has focused on the forecast danger level (e.g. Elder and Armstrong, 1987; Giraud et al., 1987; Brabec and Stucki, 1998; Lazar et al., 2016; Techel and Schweizer, 2017; Techel et al., 2018) and more rarely on other forecast elements, such as the avalanche problem (Statham et al., 2018b). Little research has explored either consistency or quality of symbols or the text used by forecasters when assessing avalanche danger. For instance, when asking avalanche professionals to assign a probability to the meaning of the five classes describing the likelihood of avalanches (Thumlert et al., 2020), or to rate the size of observed avalanches (Moner et al., 2013), considerable variation in responses was noted. Most studies have treated the forecast as a product and, for example, explored usability by testing whether users were aware of different elements of the forecast (Winkler and Techel, 2014) or whether users understood the information presented (LWD Steiermark, 2015). Engeset et al. (2018, in Norway) tested the comprehension of text, symbols and pictures, and noted that ability to comprehend the information provided in the forecast depends on the competency of the user and the complexity of the avalanche scenario. Recent work in North America used interviews to develop a typology of avalanche forecast users' competency with respect to forecast content (St. Clair et al., 2021). In a similar vein, Finn (2020) surveyed the literacy of forecast readers with respect to standardized terms and icons used in North America (such as listing or ordering the danger levels, identifying avalanche problem icons, or applying information in a slope-choice experiment). Clark (2019) explored the characterization of the severity of the avalanche problem in Canadian avalanche forecasts, described by the likelihood

of avalanches (comparable to the combination of snowpack stability and the frequency distribution of snowpack stability in Europe) and avalanche size for each avalanche problem type separately, in relation to the avalanche danger rating. Clark (2019) noted variations in the way the same avalanche danger level was characterized between different avalanche problems.

However, to our knowledge, there has been no systematic exploration of how forecasters actually describe the avalanche conditions in the narrative part of avalanche forecasts, despite such textual descriptions being the main way of communicating the factors describing the severity of the expected avalanche conditions in many forecast products in Europe (Tab. 1).

Standards used in avalanche forecasting, including the avalanche danger scale (Tab. 2; EAWS, 2018), the avalanche problems (EAWS, 2021b), the avalanche size classification (EAWS, 2019), or the conceptual model of avalanche hazard (Statham et al., 2018a), make use of specific terms to describe the stability of the snowpack (or what it takes to trigger an avalanche), the frequency distribution of snowpack stability (or how frequent triggering spots are), and what the expected avalanche size and hence the damage potential is. Nonetheless, these descriptions often include undefined, ambiguous or hedged statements (Schweizer et al., 2020; Ebert and Milne, 2021) allowing considerable room for interpretation by both forecasters and forecast users. Furthermore, there are many possible ways for forecasters to incorporate individual terms into the narrative texts often used to communicate details of avalanche danger in Europe (Tab. 1).

Thus, to answer the question: How is avalanche danger described in textual descriptions in avalanche forecasts?, and focusing on the perspective of the forecaster, leads to our first research question: (1) How well do forecasters agree on the meaning of terms characterizing triggers required to release avalanches, frequencies of triggering spots, and expected avalanche sizes? Our second research question builds on this characterization, and asks: (2) How does the use of language in danger descriptions relate to avalanche danger? To address these questions, we explored danger descriptions, written using a controlled-language environment (referred to as the *catalogue of phrases*, Winkler and Kuhn, 2017), and avalanche danger published in more than 1,000 avalanche forecasts by the national avalanche warning service in Switzerland during eight forecast seasons. The use of the catalogue of phrases is important, as it impacts the forecast product since all forecasters use the same set of words. It also impacts the analysis as the number of possible combinations of words and phrases, though large, is finite and known. We analyzed these texts in a two-step process: first, starting from an iterative annotation process (Pustejovsky and Stubbs, 2013), we annotated textual phrases relating to avalanche danger for further analysis. In a second step, Swiss avalanche forecasters related these text phrases directly to contributing factors of avalanche danger (RQ1). Building on this classification, we then extracted and analyzed text used in published danger descriptions and related it to the reported avalanche danger levels (RQ2).

## 2 Public avalanche forecasts in Switzerland

In Switzerland, the national avalanche warning service at WSL Institute for Snow and Avalanche Research SLF, Davos is responsible for the publication of avalanche forecasts covering the Swiss Alps and the Jura mountains.

Avalanche forecasters in Switzerland use definitions and guidelines provided by the European Avalanche Warning Services (EAWS) when assessing and communicating avalanche danger. Of particular relevance is the European Avalanche Danger Scale (EADS; EAWS, 2018, Tab. 2), which qualitatively describes the five danger levels (1-Low, 2-Moderate, 3-Considerable,

**Table 2.** European avalanche danger scale (EAWS, 2018).

| Danger level | Snowpack stability | Likelihood of triggering |
|---|---|---|
| 1-Low | The snowpack is well bonded and stable in general. | Triggering is generally possible only from high additional loads[**] in isolated areas of very steep, extreme terrain[**]. Only small and medium-sized natural avalanches are possible. |
| 2-Moderate | The snowpack is only moderately well bonded on some steep slopes[*]; otherwise well bonded in general. | Triggering is possible primarily from high additional loads[**], particularly on the indicated steep slopes[*]. Very large natural avalanches are unlikely. |
| 3-Considerable | The snowpack is moderately to poorly bonded on many steep slopes[*]. | Triggering is possible even from low additional loads[**] particularly on the indicated steep slopes[*]. In certain situations some large, in isolated cases very large natural avalanches are possible. |
| 4-High | The snowpack is poorly bonded on most steep slopes[*]. | Triggering is likely even by low additional loads[**] on many steep slopes[*]. In some cases, numerous large and often very large natural avalanches can be expected. |
| 5-Very High | The snowpack is poorly bonded and largely unstable in general. | Numerous very large and often extremely large natural avalanches can be expected, even in moderately steep terrain[*]. |

[*] The avalanche-prone locations are described in greater detail in the avalanche forecast (altitude, slope aspect, type of terrain): *moderately steep terrain*: slopes shallower than about 30 degrees; *steep slopes*: slopes steeper than about 30 degrees; *very steep, extreme terrain*: particularly adverse terrain related to slope angle (more than about 40 degrees), terrain profile, proximity to ridge, smoothness of underlying ground surface.

[**] Additional loads: *low*: individual skier / snowboarder, riding softly, not falling; snowshoer; group with good spacing (minimum 10 m) keeping distances. *high*: two or more skiers / snowboarders etc. without good spacing (or without intervals); snowmachine; explosives. *natural*: without human influence.

4-High, 5-Very High) in terms of snowpack stability and the likelihood of triggering. The EADS links triggers typically required to release avalanches, the number of potential triggering spots or of avalanches, the probability of avalanche release, and the potential size of avalanches to the ordinal danger levels.

Swiss avalanche forecasts describe expected regional avalanche conditions, always communicating the highest danger level
expected during the forecast period (SLF, 2020). In Switzerland dry-snow conditions, where they exist, are always summarized by a danger rating, with the danger description describing these dry-snow conditions and relevant avalanche problems. Wet-snow conditions, on the other hand, are often mentioned only as a secondary problem. This in turn means that danger resulting from secondary wet-snow problems is at most as high as the danger level communicated with the primary problem, but may often be lower. A danger rating referring to wet-snow conditions is only given if the danger of wet-snow or gliding avalanches
exceeds that of dry-snow avalanches.

To communicate spatial variation in avalanche conditions, the forecast area is divided into almost 150 *warning regions*, the smallest spatial units used in the forecast. These warning regions are aggregated flexibly to *danger regions*, which are

characterized with the same danger level, avalanche problems, avalanche-prone locations and danger description (see example in Fig. 1b, which shows the forecast components describing avalanche danger in a danger region). Each danger region contains a textual description of avalanche conditions. In Switzerland, the severity of the avalanche situation is described exclusively in the danger description (see also Tab. 1 for a comparison with other forecasts). The trigger, the frequency of triggering spots and avalanche size described in the text may refer to a specific avalanche problem (as in the example in Fig. 1b) or to the avalanche conditions as a whole.

Avalanche forecasts are published in the evening at 1700 CET[1] and are valid until the following day at 1700 CET. During the main winter season forecasts are also updated at 0800 CET, remaining valid until 1700 CET on the same day.

Since November 2012, the text of Swiss avalanche forecasts has been prepared using a controlled-language environment, relying on a so-called *catalogue of phrases* (Winkler et al., 2013; Winkler and Kuhn, 2017) - a collection of predefined sentence templates, permitting instantaneous, automatic translation of the German text into French, Italian and English. The aim of such a catalogue of phrases is, for one base language, to allow generation of naturalistic texts using a restricted lexicon, base syntax and semantics Kuhn (2014). The resulting texts are understandable by speakers of the base language, and as shown by Winkler and Kuhn (2017), speakers could not distinguish between danger descriptions written using the catalogue of phrases and freely written danger descriptions. Using this catalogue of phrases also makes automatic translation straightforward, and recently the catalogue has been extended to Spanish and Catalan (since winter 2020/21).

The catalogue of phrases consists of a number of *sentences*, each containing a number of *phrase_options*. Phrase_options contain either *values*, the actual textual content, or up to two additional levels of phrase_options, thus allowing an enormous number of possible combinations. An example, illustrating the creation of a single sentence, is shown and explained in Figure 2.

## 3 Data

We analyzed 1,286 map-based avalanche forecasts published at 1700 CET during eight winter seasons between 27 Nov 2012 and 13 Feb 2020. 5,897 danger regions were described by a danger level, an avalanche problem, aspects and elevations where the danger prevails, and a danger description (SLF, 2020, see also example shown in Fig. 1b).

For this study, we extracted the forecast danger level (element 1 in Fig. 1), and the respective danger description related to the major problem being either dry-snow conditions or wet-snow/gliding avalanches (elements 4 and 3 in Fig. 1, Tab. 3).

## 4 Methods

We worked with German danger descriptions, as German is both the working language of forecasters in Switzerland and the base language used in the creation of the catalogue of phrases. Since the danger descriptions, written with the catalogue of phrases, are a form of controlled natural language (Kuhn, 2014), they can be analysed using standard natural language

---
[1]CET or CEST, respectively

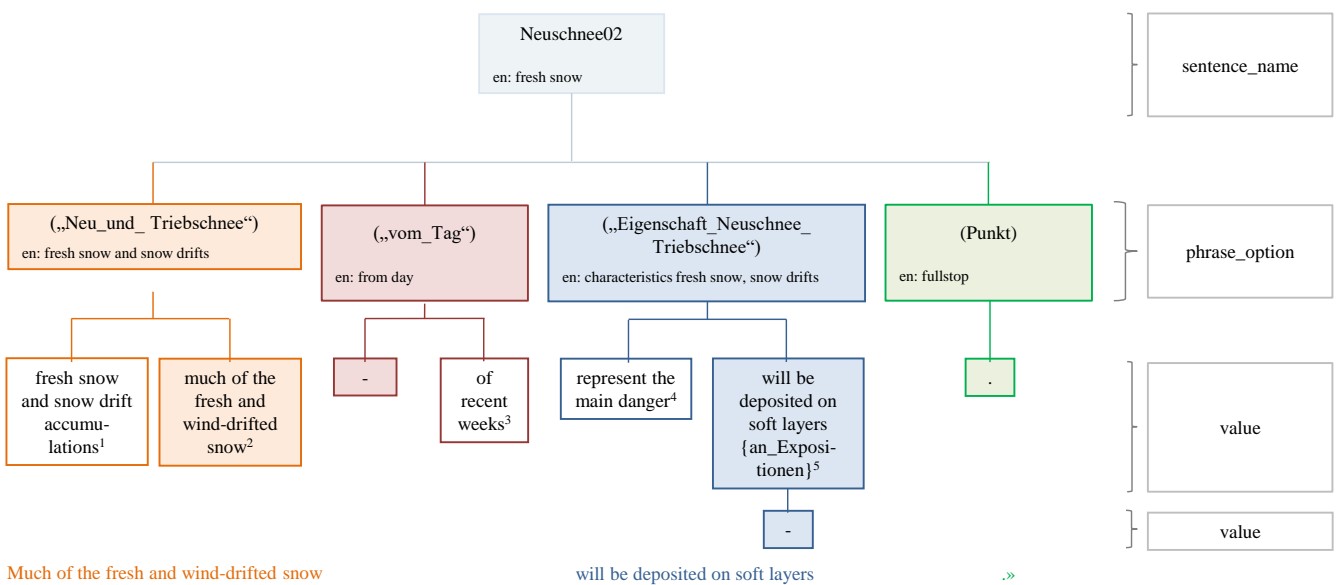

German original *values*

[1] Neu- und Triebschnee | [2] Viel Neuschnee und Triebschnee | [3] der letzten Wochen | [4] bilden die Hauptgefahr | [5] werden {an_Expositionen} auf weiche Schichten abgelagert

**Figure 2.** Structure of the catalogue of phrases. A *sentence* consists of *phrase_options*. These must be filled with *values*. In this example, the sentence labeled *Neuschnee02* is selected. This sentence contains four phrase_options, each providing a set of values (text modules). In this example, only two of the possible values are shown. The operational language in the Swiss catalogue of phrases is German.

**Table 3.** Data overview: avalanche forecasts.

| type | 1-Low | 2-Mod | 3-Cons | 4-High | 5-vHigh | all |
|---|---|---|---|---|---|---|
| dry-snow | 1031 | 2245 | 1836 | 158 | 4 | 5274 |
| wet-snow | 177 | 300 | 133 | 13 | 0 | 623 |
| all | 1208 | 2545 | 1969 | 171 | 4 | 5897 |

2-Mod: 2-Moderate; 3-Cons: 3-Considerable; 5-vHigh: 5-very High

processing methods. As we are working with a finite corpus of phrases, and we are interested in how those meanings or phrases can be and are interpreted (the semantics), we manually annotated textual elements in the catalogue of phrases (Pustejovsky and Stubbs, 2013). We followed the iterative annotation proposed by Pustejovsky and Stubbs (2013) which they describe as the (Model-Annotate-Model-Annotate) MAMA cycle.

Our starting point was the catalogue of phrases and its 9'989 unique values. As only some values contain information related to the three key factors characterizing avalanche danger - type of trigger, the frequency of triggering spots and avalanche size, a two-step approach was used to annotate the text.

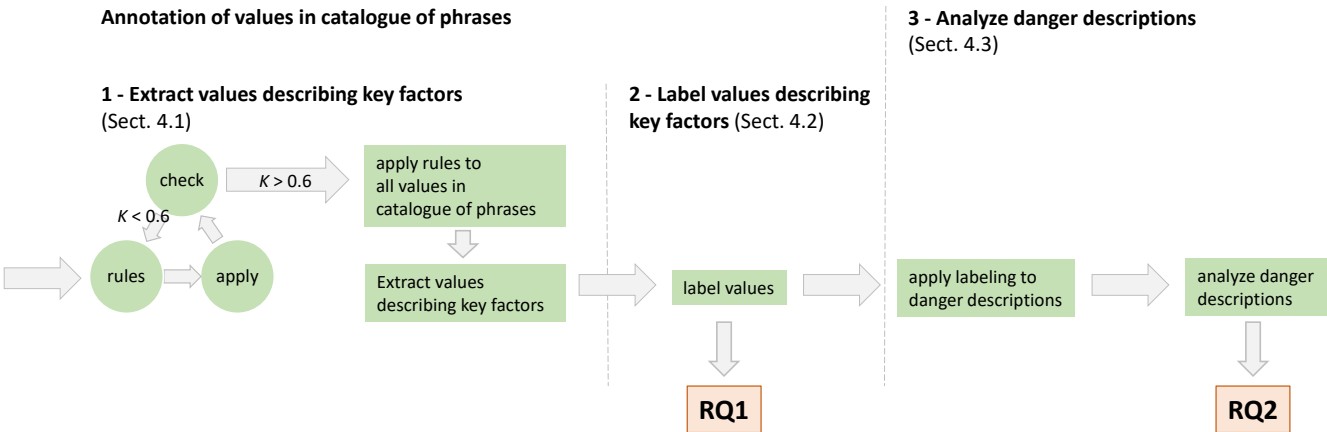

**Figure 3.** Workflow describing the annotation of values in the catalogue of phrases and the analysis of the danger descriptions. The annotation (step 1) and labeling (step 2) of the values allowed to answer research question 1 (RQ1: «How well do forecasters agree on the meaning of terms characterizing triggers required to release avalanches, frequencies of triggering spots, and expected avalanche sizes?»), while the application of the labeling to the danger descriptions (step 3) permitted to answer RQ2 («How does the use of language in danger descriptions relate to avalanche danger?»). For details refer to text.

First, rules were defined (step 1 in Fig. 3), before individual values in the catalogue of phrases were assigned to one of the three key factors in an iterative annotation process using these rules (Sect. 4.1). This initial step allowed us to retain only values contained in the catalogue of phrases judged to describe one of the three key factors, considerably reducing the amount of labeling to be done in the following step. Second, (step 2 in Fig. 3), the values which characterize the key factors were assigned ordinally-ranked thematic labels according to current practice and/or suggested labeling used in avalanche forecasting or recent
research (Sect. 4.2). These two steps allowed us to answer research question 1: «How well do forecasters agree on the meaning of terms characterizing triggers required to release avalanches, frequencies of triggering spots, and expected avalanche sizes?»

Finally, danger descriptions and the labels assigned to them were analysed with respect to avalanche danger (step 3 in Fig. 3, Sect. 4.3) permitting the exploration of research question 2: «How does the use of language in danger descriptions relate to avalanche danger?»

## 4.1 Catalogue of phrases: annotation and extraction of values describing key factors

The values in the catalogue of phrases were labeled in an iterative process (c.f. Hutter (2020)), summarized as follows (see also Fig. 3 - step 1):

– (a) Annotation rules were developed based on the definitions or descriptions of key factors in scientific literature and operational guidelines. We distinguished three key factors which we refer to as the *trigger type*, the *frequency of triggering*
*spots*, and *avalanche size* (Tab. 4). A short version of the annotation rules is provided as a supplement in the Appendix A1. In addition, we also annotated text values specifying the *location of triggering spots*.

**Table 4.** Labels and number of categories used to describe the key factors. Additionally to the labels shown, a value could also be labeled *not assignable*. The order of the labels corresponds to the rank order used in this analysis with the left-most labels representing the most unfavorable conditions and the right-most label the most favorable conditions. A comparison with the terms used in the conceptual model of avalanche hazard (CMAH) is also provided as a guide.

| key factor | categories | labels | key factor (CMAH) |
|---|---|---|---|
| trigger type | 2 | natural (expected, possible), additional load (low, high) | sensitivity to triggers |
| frequency of triggering spots | 3 | many, some, a few | spatial distribution |
| location of triggering spots | | text elements, not labelled (see also Tab. A5) | |
| avalanche size | 5 | 5 - extremely large, 4 - very large, 3 - large, 2 - medium, 1 - small | avalanche size |

- (b) Relying on these rules, all the values in a set of ten randomly selected danger descriptions (two for each danger level) were annotated with regard to which of the key factors they belonged to.

- (c) Following annotation, the agreement between pairs of annotators was assessed by calculating the inter-rater agreement score (Cohen's kappa coefficient $\kappa$), which also takes into account the agreement by random chance (Landis and Koch, 1977).

- (d) Steps (a) to (c) were repeated until sufficient agreement in labeling was achieved. We considered a sufficient agreement if the minimal agreement between any of the annotators was $\kappa > 0.6$, which is considered *substantial* agreement according to Landis and Koch (1977). To achieve this level of agreement, three annotation rounds, after each of which the annotation rules were discussed and revised, were carried out. In each of these rounds, ten new danger descriptions were annotated. The agreement between annotators increased from $\kappa > 0.54$ (round 1) to $\kappa > 0.7$ (round 3).

- (e) Applying the annotation rules, the values contained in the catalogue of phrases were assigned to a key factor. As only a small subset was annotated in the three annotation rounds (5% of the values), the assigned labels were quality-checked resulting in the inclusion of two additional values. About 1200 values contained information characterizing one of the three key factors.

## 4.2 Catalogue of phrases: labeling of values

Once the values were assigned to key factors, the second step was the labeling of the individual values (see also Fig. 3 - step 2).

We first grouped values with very similar meanings. Values considered similar included variations which we judged to be synonymous such as «even in places that are not usually affected» [2] and «even in places that are usually less vulnerable» [3], or singular and plural forms. This reduced the original set of 1200 values to 109.

---

[2] German original: an sonst wenig gefährdeten Orten

[3] German original: an sonst weniger gefährdeten Orten

To label the values, no further annotation rules were defined as we were interested in how the forecasters (our annotators) understood these values (RQ 1). The number and labels of classes was based on definitions and descriptions used in avalanche forecasting in Europe (Tab. 4): five avalanche size classes and their official labels (EAWS, 2019), the distinction of trigger types as natural or artificial triggers (EAWS, 2018), and three classes for the frequency of triggering spots or number of avalanches as in the current working documents of the European Avalanche Warning Services (e.g. EAWS, 2021a). In addition, two probability terms are used in Switzerland to describe the occurrence of natural avalanches.

- (i) Three annotators assigned a single class to the 109 groups of values, including the option that a class could not be assigned. The inter-rater agreement rates ranged from 0.53 (considered *moderate*) and 0.63 (considered *substantial*). 53% of the groups of values were rated the same by all three annotators. 22 of the values could not be assigned to a class by at least two of the three annotators. For instance, text describing avalanches releasing *deep within the snowpack*[4] or *weak layers existing close to the snow surface*[5], could be interpreted as being related to avalanche size. These text values were therefore assigned a relation with avalanche size in the annotation step described in the previous section (Sect. 4.1). However, in the annotation round described here, when annotators were specifically asked to assign a size class (or two), none could do so.

- (ii) Removing the values which could not be assigned to a class in the first round (i), the eight avalanche forecasters working at SLF assigned one or two classes to values. The inter-rater agreement rate $\kappa$ was on average 0.74 (considered *substantial*, Landis and Koch, 1977) between any two annotators, but ranged between 0.64 (considered *substantial*) and 0.87 (considered *almost perfect*, Fig. 4). 53% of the values were assigned the same class by all eight forecasters.

If five annotators (a majority) indicated the same class, the value was assigned to this class. If there was no clear majority vote, the value was assigned to the two most frequent classes chosen. The values and their assignment to classes are shown in the Appendix (Tab.s A2 to A4), with their German original, a corresponding English translation, and the assigned class labels. For the remainder of this manuscript, we refer purely to the class labels shown in Tab. 4.

Values which described the location of potential trigger locations were not categorised. An overview of these values is given in the Appendix (Tab. A5).

## 4.3 Danger description: analysis

Applying the annotated catalogue of phrases to the actual danger descriptions (Fig. 3 - step 3), we were able to analyze the content of the danger descriptions. Labels were assigned to values according to Tab.s A2 to A4. For example, as shown in Fig. 5, the values *reach large size* (= size 3), *very isolated cases* (= a few) and *can be triggered* (= additional load) would be used for further analysis. Where a value was not linked to a single class, we randomly selected one of the two most frequent labels rather than removing these cases or always opting for a more unfavorable label. This random assignment was primarily

---

[4]German original: tief in der Schneedecke

[5]German original: Schwachschichten nahe an der Schneeoberfläche

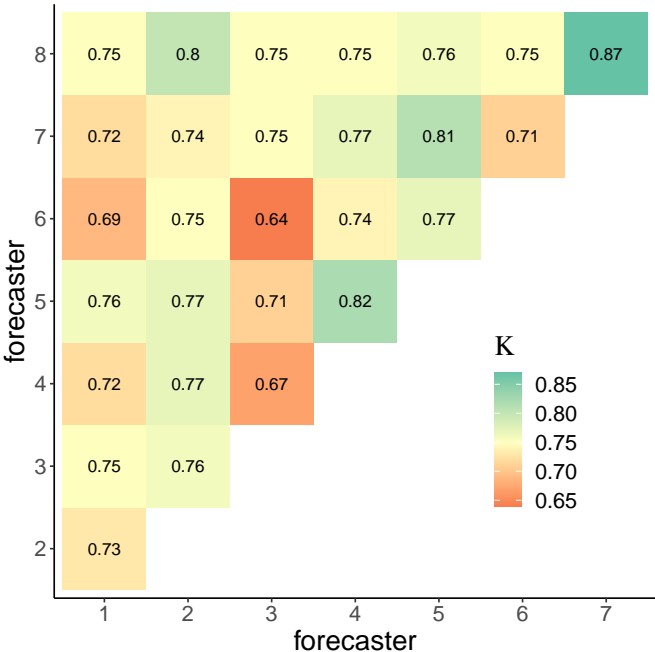

**Figure 4.** Kappa scores for the eight forecasters (f1-f8). $\kappa > 0.6$ is considered *substantial*, $\kappa > 0.8$ *almost perfect* (Landis and Koch, 1977).

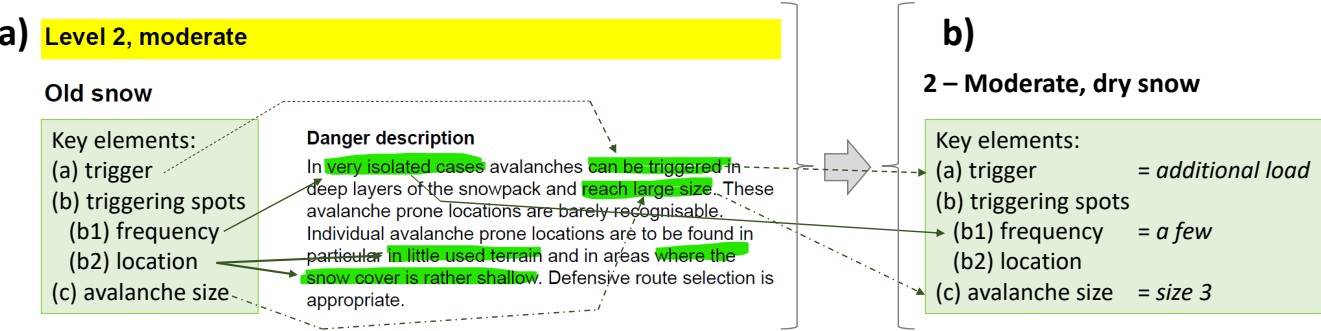

**Figure 5.** Following the annotation and labeling (Sect.s 4.1 and 4.2), the key factors and their labels were extracted from each danger description. Here, the same example as used in Fig. 1b is shown.

required for values referring to avalanche sizes (cf. Tab. A3), as some terms were assigned to two size classes. For instance, *rather large*[6] was linked to avalanche sizes 2 and 3 by seven of the eight forecasters.

We analyzed danger descriptions relating to dry-snow and wet-snow conditions separately. We make this distinction as
the danger rating and the accompanying danger description often refers to dry-snow conditions in the Swiss forecast (Sect. 2). In addition, wet-snow avalanches almost always release naturally (e.g. Schweizer et al., 2020), in contrast to dry-snow

---
[6]German original: ziemlich gross, cf. Tab. A3

avalanches, where natural released and artificially triggering avalanches are of equal concern. The EADS, however, does not make this distinction between dry-snow and wet-snow conditions.

We conducted the analysis in two steps: First, we explored whether information describing each of the key factors was present in the danger descriptions, regardless of their label. Second, we analyzed the frequency that a certain class was mentioned, considering all danger descriptions of a specific data subset (e.g. a specific danger level). To do so, we searched for the most unfavorable piece of information describing a specific key factor within a danger description relying on the rank-order shown in Table 4. To compare two proportions, we relied on a one-sided proportion test (R-function *prop.test*, Newcombe, 1998), testing the hypothesis as to whether the proportion in one subgroup was significantly lower (or higher) than in another subgroup. We always explored the proportions from sequential danger levels (i.e. for 2-Moderate and 3-Considerable) in a pair-wise fashion. We refer to results as statistically significant if $p < 0.05$ and report $p$-values in three classes: $p < 0.05$, $p < 0.01$, $p < 0.001$. Where the proportion test indicated significant differences for comparisons of consecutive danger level pairs, we simply state the $p$-value class indicating the lowest significance (the highest value for $p$).

The entire analysis was conducted using the software *R* (R Core Team, 2020).

## 5 Results

### 5.1 Description of dry-snow avalanche conditions

5'274 danger descriptions referred to dry-snow conditions (Tab. 3). Of these, avalanche size (proportion 0.68 of all cases) and the frequency of triggering spots (referred to as frequency, 0.76) were described most of the time, while information on the type of trigger was provided as often as not (0.53). In addition to describing the frequency, 0.72 of the danger descriptions specified the location of triggering spots (referred to as location). Text indicating either a frequency or a specific location was indicated in 0.9 of the danger descriptions.

The proportion of danger descriptions providing information on all three elements characterizing avalanche danger decreased consistently from one danger level to the next lower one (see also uppermost row labelled *all 3\** factors in Tab. 5). Differences were significant ($p < 0.001$) for all comparisons, except when comparing 5-Very High (0.75) and 4-High (0.52). The description of 2-Moderate was the most incomplete in this regard: 0.34 of the danger descriptions described only one or none of the three key factors (cf. Tab. 5, rows labelled *1 of 3\** or *none\**).

In the following, we explore the actual values of the key factors in the danger descriptions for each danger level. The results are summarized in Figure 6a, c, and e, and Table 6.

The proportion of danger descriptions, which indicated a trigger, increased clearly with increasing danger level (Fig. 6a). A trigger was rarely specified at 1-Low (0.03), more often at 2-Moderate (0.43), and most of the time or always at the other danger levels ($\geq 0.88$). We labeled the trigger required to release an avalanche as either a *natural avalanche* or requiring an *additional load* (cf. Tab. 4). All the danger descriptions at 4-High and 5-Very High indicated natural avalanches, compared to a proportion of 0.33 at 3-Considerable, 0.02 at 2-Moderate, and 0.005 at 1-Low (Fig. 6a). Overall, the proportion of danger descriptions, which mentioned natural avalanche occurrence increased significantly from one danger level to the next higher for all danger

**Table 5.** Proportion of dry-snow danger descriptions which contained information on the key factors (trigger type, frequency and location of triggering spots, avalanche size). The figure provides two levels of detail. First, the proportions that information on the three key factors characterizing avalanche danger, marked with an *, was given in the danger description (i.e. when **2 of 3*** factors were described). These proportions are highlighted bold. Second, proportions are shown for each individual combination of factors within these sub-groups, with a 1 indicating when a key factor was described and 0 when a key factor not described (i.e. in the second row, when *trigger*, *frequency* and *avalanche size* were described (= 1), but not *location* (= 0)).

| factors* | trigger type* | triggering spots frequency* | location | avalanche size* | 1-Low N = 1031 | 2-Mod N = 2245 | 3-Cons N = 1836 | 4-High N = 158 | 5-vHigh N = 4 |
|---|---|---|---|---|---|---|---|---|---|
| **all 3*** | | | | | **0.03** | **0.21** | **0.36** | **0.52** | **0.75** |
| | 1 | 1 | 0 | 1 | 0 | 0.05 | 0.15 | 0.28 | 0.5 |
| | 1 | 1 | 1 | 1 | 0.03 | 0.16 | 0.2 | 0.23 | 0.25 |
| **2 of 3*** | | | | | **0.82** | **0.45** | **0.46** | **0.47** | **0.25** |
| | 0 | 1 | 1 | 1 | 0.78 | 0.2 | 0.03 | 0 | 0 |
| | 1 | 0 | 0 | 1 | 0 | 0.01 | 0.18 | 0.19 | 0.25 |
| | 1 | 0 | 1 | 1 | 0 | 0.04 | 0.14 | 0.28 | 0 |
| | 1 | 1 | 1 | 0 | 0 | 0.11 | 0.06 | 0 | 0 |
| | 0 | 1 | 0 | 1 | 0.03 | 0.05 | 0.01 | 0 | 0 |
| | 1 | 1 | 0 | 0 | 0 | 0.04 | 0.04 | 0 | 0 |
| **1 of 3*** | | | | | **0.14** | **0.3** | **0.15** | **0.01** | **0** |
| | 0 | 1 | 1 | 0 | 0.11 | 0.19 | 0.02 | 0 | 0 |
| | 1 | 0 | 0 | 1 | 0 | 0.02 | 0.06 | 0 | 0 |
| | 0 | 1 | 0 | 0 | 0 | 0.06 | 0.01 | 0 | 0 |
| | 0 | 0 | 1 | 1 | 0.02 | 0.02 | 0.01 | 0 | 0 |
| | 1 | 0 | 0 | 0 | 0 | 0 | 0.05 | 0 | 0 |
| | 0 | 0 | 0 | 1 | 0.01 | 0 | 0 | 0 | 0 |
| **none*** | | | | | **0.01** | **0.04** | **0.04** | **0** | **0** |
| | 0 | 0 | 1 | 0 | 0 | 0.03 | 0.02 | 0 | 0 |
| | 0 | 0 | 0 | 0 | 0 | 0.01 | 0.01 | 0 | 0 |

level pairs ($p < 0.001$). In the Swiss forecast, two German terms are used to describe the probability of natural avalanche release: *expected* or *probable*[7] indicating a high probability, and *possible*[8] indicating a lower probability. For cases, when either of these terms was used, it was generally *expected* or *probable* at 4-High and 5-Very High (0.88 and 1.0, respectively), and mostly *possible* at the other danger levels (1-Low 0.96, 2-Moderate 0.9, 3-Considerable 0.76). In contrast, an additional load was comparably rarely indicated; at 1-Low (0.03 of the time), and about as often as not at 2-Moderate (0.41) and 3-Considerable (0.55, cf. Fig. 6a). If an additional load was specified, it was mostly described as a low additional load at 3-Considerable (0.98) and at 2-Moderate (0.68), and a high additional load for the few cases containing this information at 1-Low (0.72).

---

[7]German original: zu erwarten
[8]German original: möglich

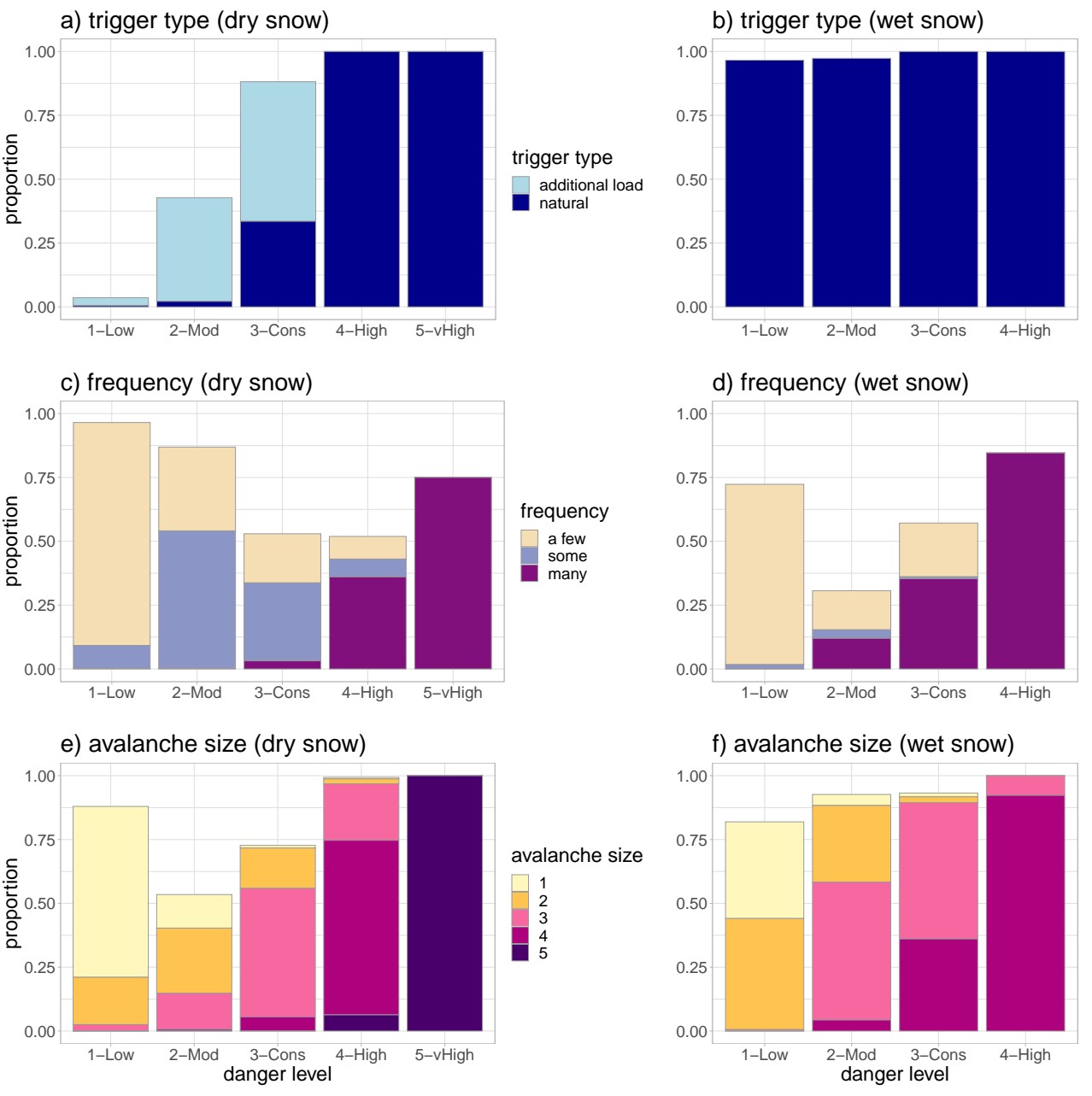

**Figure 6.** Barplots showing the proportions for the terms characterizing the three key factors by danger level. The left column (a, c, e) shows the distributions for dry-snow conditions, and the right column (b, d, f) for wet-snow conditions. The upper row (a, b) shows the trigger, the middle row (c, d) the frequency of triggering spots, and the lower row (e, f) the avalanche size. No data for wet-snow conditions for danger level 5-Very High.

**Table 6.** Description of the contributing factors of avalanche danger for dry-snow and for wet-snow conditions, and - as comparison - in the European Avalanche Danger Scale (EADS, extracted from Tab. 2; EAWS, 2018). Results from this study are summarized by showing the list of the most frequent terms ordered according to their frequency of occurrence, and if more than 20% of the time. The words in the EADS are translated according to the annotation in Tables A2-A4 and the overview by Müller et al. (2016). *n.d.* indicates that a key factor was not described in the danger descriptions or the EADS. If *n.d.* is listed first, this means that this key factor was more often not described compared to the most frequent label which was described. – no data for wet-snow conditions at 5-Very High.

| danger level | factor | dry | wet | EADS |
|---|---|---|---|---|
| 1-Low $N = 1208$ | trigger type / probability | n.d. | natural possible | high additional load |
| | frequency | a few | a few, n.d. | a few |
| | avalanche size | 1 | 2, 1 | 1 and 2 |
| 2-Moderate $N = 2545$ | trigger type / probability | n.d., additional load | natural possible | high additional load |
| | frequency | some, a few | n.d. | some, specific |
| | avalanche size | n.d., 2 | 3, 2 | n.d., $< 4$ |
| 3-Considerable $N = 1969$ | trigger type / probability | additional load, natural | natural expected | low additional load |
| | frequency | n.d., some | n.d., many, a few | some, many |
| | avalanche size | 3, n.d. | 3, 4 | $\leq$ 3 - 4 |
| 4-High $N = 171$ | trigger type / probability | natural expected | natural expected | natural expected, low additional load likely |
| | frequency | n.d., many | many | many |
| | avalanche size | 4, 3 | 4 | 3 and 4 |
| 5-Very High $N = 4$ | trigger type / probability | natural expected | – | natural expected |
| | frequency | many, n.d. | – | many |
| | avalanche size | 5 | – | 4 and 5 |

The frequency of potential triggering spots or of the number of avalanches, classified as either *a few*, *some* or *many*, was described about half of the time (proportion 0.5) at 3-Considerable and 4-High, and more often at the other danger levels ($\geq$ 0.75, Fig. 6c). When the frequency was described, it was most often *a few* at 1-Low, *some* at 2-Moderate and 3-Considerable, and *many* at 4-High and 5-Very High (Fig. 6c, Tab. 6). The proportion of danger descriptions which indicated *a few* locations decreased significantly ($p < 0.001$) from 1-Low (0.87) to 2-Moderate (0.33), from 2-Moderate to 3-Considerable (0.19), and from 3-Considerable to 4-High (0.09). Similarly, the proportion of forecasts which mentioned *many* locations, increased significantly ($p < 0.001$) from 2-Moderate (0.003) to 3-Considerable (0.03), and from 3-Considerable to 4-High (0.36).

Beside simply describing the frequency of potential triggering spots, a specific description of where in the terrain these spots were likely to be was provided often at 1-Low (0.95) and 2-Moderate (0.8), less often at 3-Considerable (0.57) and 4-High (0.54), and rather seldom at 5-Very High (0.25). In other words, pointing out specific locations was more often the case at

danger levels when the frequency was rather low (*a few* or *some*). In most danger descriptions, there was at least an indication of either the frequency or the location of triggering spots (between 0.75 at 5-Very High and 0.99 at 1-Low).

Avalanche size was comparably rarely indicated at 2-Moderate (0.53 of cases, Fig. 6e). For cases, when an avalanche size was indicated, there was a perfect monotonic correlation between the most frequently indicated avalanche size and the danger level, with, for instance, size 1 being most frequently indicated at 1-Low and size 5 at 5-Very High (Fig. 6e and Tab. 6). As outlined in Section 4.3, we opted to randomly assigning labels where forecasters had no majority opinion with regard to the classification of textual elements. This was particularly common for some frequently used terms describing avalanche size. This meant that 35% of the textual elements were therefore randomly assigned to avalanche size 1 or 2, or size 2 or 3. If we had instead consistently assigned the larger of the avalanche sizes, at 1-Low the proportions of size 2 avalanches would have increased (from 0.19 to 0.33) at the cost of size 1 avalanches (0.67 to 0.52). A similar, though less pronounced shift in the proportion of size 3 avalanches would be observed at 2-Moderate (from 0.14 to 0.17) and 3-Considerable (from 0.5 to 0.57). However, correlation between the most frequently indicated avalanche size and the danger level would still be perfectly monotonic.

## 5.2 Description of wet-snow or gliding avalanche conditions

Summaries regarding the description of wet-snow or gliding avalanche conditions can be found in Figure 6b, d and f, and Table 6.

623 danger descriptions described wet-snow or gliding avalanches as the primary danger (Tab. 3). These were almost always described as natural avalanches ($> 0.96$, Fig. 6b). The probability terms used to describe the occurrence of natural avalanches were predominantly *possible* at 1-Low (0.94), about as often *possible* (0.49) or *expected* (0.51) at 2-Moderate, while at 3-Considerable and 4-High natural avalanches were almost always *expected* (0.93 and 1.0, respectively).

The description of the frequency of expected avalanches showed a bi-modal distribution with the middle class *some* rarely being used (0.05). Furthermore, frequency information was missing in 0.7 of the cases at 2-Moderate, and 0.45 of the time at 3-Considerable (Fig. 6d). However, as for dry-snow conditions, in these cases an indication of the location of potential release areas was often made in the text. When frequency information was indicated, it was essentially always *a few* at 1-Low (0.98) and always *many* at 4-High. Considering text information describing the frequency and location of release spots together, the danger descriptions contained at least one piece of information in this regard (between 0.73 at 2-Moderate and 0.92 at 1-Low).

Avalanche size was often indicated ($> 0.8$ of the time), and increased from size 1 to 2 at 1-Low (0.46 and 0.53, respectively) to size 2 to 3 at 2-Moderate (0.32 and 0.58, respectively), to size 3 to 4 at 3-Considerable (0.57 and 0.39, respectively), to size 4 at 4-High (0.92, cf. Fig. 6f).

## 6 Discussion

Our approach aimed to better understand how words and phrases from a structured catalogue are used to convey avalanche danger. In contrast to the relatively small number of other studies which have concerned themselves with the communication

of avalanche danger through forecasts (e.g. Burkeljca, 2013; Engeset et al., 2018; St. Clair et al., 2021; Finn, 2020), our starting point was to explore how forecasters interpret (RQ1) and use narrative text to convey avalanche hazard (RQ2). We took advantage of a unique dataset to perform our analysis: avalanche forecasts written over eight winter seasons using a structured catalogue in Switzerland. To discuss our results, we introduce here the notion of the semiotic triangle (Ogden and Richards, 1925; MacEachren, 2004), a concept commonly used in linguistics and cartography to understand the relationships between a

- *referent*, an instance in the real world, in our case a (partially observable) avalanche situation,

- *thought*, the mental models used to form a judgement about a situation, and

- *symbols*, the words or icons used to describe a referent.

This triangle is helpful as it emphasises that the process of moving from a referent (the avalanche situation) to a symbol (the avalanche forecast) is influenced by those observing and communicating that situation and that this process is not completely knowable. Perhaps the most important aspect of the semiotic triangle with respect to forecasting is that it makes explicit the obvious, but often forgotten, reality that a forecast is an abstraction of a reality, understood by individuals, and communicated through symbols.

To answer the first research question, »How well do forecasters agree on the meaning of terms characterizing triggers required to release avalanches, frequencies of triggering spots, and expected avalanche sizes?«, we asked forecasters to assign labels to text values available in the catalogue of phrases to describe these factors (Sect. 4.2, Tab.s A2 - A4). This annotation process was a necessary step in exploring our second research question, since we needed these labels to understand how avalanche danger was described by forecasters. However, equally importantly, it gave us insight into the degree to which a trained team of forecasters used language to describe different characteristics of avalanche danger. Since the task was performed in isolation - that is to say forecasters classified terms independently of a specific avalanche situation, it relates to one side of the semiotic triangle - the relationship between the symbol (the language used to convey a situation using the varied options available in the catalogue of phrases) and thought (the abstraction of an avalanche situation described using a small number of key factor labels). Although the overall agreement in the assigned labels between forecasters was rather high (cf. Fig. 4, $\kappa >$ 0.67), with 50% of the text symbols being assigned to the same class by all forecasters, it is important to note that these values are based on expert annotation by a team working together on a daily basis.

Zooming into the individual classifications, it is possible to identify areas for discussion in the forecasting team with regard to three issues. First, the terms used most consistently were those taken directly from definitions. For example, there was 100% agreement about the use of terms used in the definition of avalanche size classes (e.g. *small*[9] avalanche for a size 1 avalanche or *very large*[10] avalanche for a size 4 avalanche). Second, other terms, especially those which hedged, were considered more ambiguous by the forecasters, with for example *rather small*[11] avalanches being considered by 4 forecasters as representative

---

[9]German original: klein
[10]German original: sehr gross
[11]German original: ziemlich klein

of size class 1 and 4 forecasters of size class 1 – 2 avalanches. This difference matters since size 1 avalanches are typically not associated with burials, while size 2 may «bury, injure or kill a person» (EAWS, 2019), and as size 2 avalanches more often lead to burials of recreationalists in Switzerland. Third, we also identified a number of terms present in the structured catalogue which were never used by the forecasters. In general, annotating and assigning words and phrases to particular situations gave valuable insights into the ways in which avalanche forecasters describe avalanche situations, and help identify areas where consistency could be improved.

Our second research question asked »How does the use of language in danger descriptions relate to avalanche danger?» Answering it provides us with knowledge as to how forecasters take a referent, in this case the expected evolution of the avalanche situation over the next 24 hours, and represent it through language. The annotations of words and phrases used in the avalanche forecast allow us to first characterize how avalanche danger is described, and second explore the consistency of descriptions of similar avalanche danger.

The description of the three elements characterizing the danger level - the trigger required to release an avalanche, the frequency of triggering spots, and the expected avalanche size, varied in their degree of completeness. Danger level 2-Moderate avalanche danger in dry-snow conditions was characterized by all three factors only 21% of the time, and 30% of descriptions only mentioned one factor (most often the frequency and location of the likely triggering spots) (Tab. 5, row *1 of 3\**). Since in Switzerland many avalanche accidents happen at this level of forecast avalanche danger (e.g. Winkler et al., 2021), characterising the likely consequences and triggers of these avalanches more often may be useful in communicating the situation. For danger levels 3-Considerable and 4-High, the frequency of triggering spots was missing about half the time (Fig. 6c).

These distributions of missing information are clearly not random, and reflect systematic choices made by the forecasters in translating the avalanche danger (referent) to a danger description (symbol) through a thought process unknown to forecast users. It appears that the cases where information is missing are those where values would likely describe the middle ground rather than the extremes (Tab. 6). Since this middle ground may in practice be where the interpretation of avalanche forecasts is more difficult for a recreationalist, and given that avalanche danger definitions include all three factors at all levels of avalanche danger, it is important to consider further the likely influence of missing information on users.

Irrespective of whether factors are described in a forecast, it is also important that the factors included are used consistently. In general, we found this to be the case and the description of the elements characterizing avalanche danger changed significantly from one danger level to the next (Fig. 6). As shown in Sect. 5 (cf. Tab. 6), dry-snow and wet-snow avalanche conditions were described differently: natural avalanches are essentially always mentioned in danger descriptions describing wet-snow or gliding avalanches, regardless of danger level (Fig. 6b), while in dry-snow conditions primarily at 3-Considerable or higher danger levels (Fig. 6a). Differences also exist regarding the largest expected avalanche size: these tended to be larger for wet-snow than for dry-snow avalanche conditions (cf. Tab. 6). For instance, for cases when avalanche size was described, size 3 avalanches were the most frequently expected avalanche size at 2-Moderate in wet-snow conditions (55%), and at 3-Considerable in dry-snow conditions (50%, Fig. 6e, f). These findings vary from the definitions given by the EADS (Tab. 6), which does not distinguish between avalanche sizes expected in dry or wet-snow conditions. They do, however, correspond well with a study exploring a large data set of avalanche occurrence data in the region of Davos (Eastern Swiss Alps), which

showed that the largest observed avalanche was larger and that the number of natural avalanches was higher for wet-snow avalanches compared to dry-snow avalanches on days with the same forecast danger level (Schweizer et al., 2020). Although this means that the description of the forecast corresponds to observations, it also highlights an inconsistency in the application of the danger levels allowing more natural avalanches at larger size in wet-snow conditions than dry-snow conditions. This may also be one explanation for variations in the use of the danger levels in Switzerland, compared to - for instance - its neighbours in Italy (Techel et al., 2018).

## 6.1 Implications for forecasters

The list of German words related to key factors and their association with a set of (categorical) labels (Appendix Tables A2 to A4) provides an opportunity to improve the consistency of the terms used to describe specific conditions by avalanche forecasters in Switzerland. This list of words may also provide a valuable starting point to harmonizing danger descriptions in other parts of the Alps, where the operating language of avalanche forecasters is German. Since hedged phrases seem to reduce consistency between forecasters, and thus cannot be correctly interpreted by forecast users, we suggest identifying and discussing the use of these terms in the catalogue of phrases.

Our results showed information is often lacking with respect to trigger type, frequency of triggering spots or avalanche size. It is unclear whether this missing information reflects a) a conscious decision by forecasters to omit information considered redundant, b) uncertainty by forecasters with respect to these factors at some danger levels or c) simply forgetting to provide information about these factors. If information is simply being forgotten, then a more structured approach, such as proposed by the CMAH would solve the problem. However, providing information for all factors presupposes that this information is reliable and relevant to users. Our results suggest that investigating the reliability and the use of these key factors in forecasts in more detail is urgent, before decisions can be made about the most effective format in which to communicate such information.

## 6.2 Implications for users of the avalanche forecast

The purpose of an avalanche forecast is, in the case of recreationalists, to provide useful information aiding decision making in planning and carrying out activities. The first requirement for a useful avalanche forecast is therefore that it is correct and consistent. Our results show that in general, the use of language to communicate and specify avalanche danger is (reasonably) consistent between forecasters and correlates with forecast avalanche danger. Here, the semiotic triangle again comes into play, as a user interprets the symbols used by a forecaster to create their own mental model of the avalanche conditions.

Our work though explored the use of language from a different perspective – that of expert forecasters. It reveals that forecasters' use of language describing avalanches situations is more consistent using words and phrases which relate directly to definitions, and that the characterisation of avalanche danger is least complete where the situation is more ambiguous. Leaving out information, for example the likely triggers or size classes of avalanches expected for danger level 2-Moderate, may, for forecasters, actually convey information about the situation. For instance, information on the type of trigger is often missing at lower danger levels. To a forecaster not mentioning natural avalanches may be a clear indication that an additional load is required to release avalanches. However, it is unlikely that users of an avalanche forecast will interpret absence of information

in this way. Again, a potential solution to this problem may be a more structured format, such as that used in Norway or in Canada (cf. Tab. 1), where important characteristics describing avalanche conditions, including expected avalanche size and whether natural avalanches are expected, are provided in tabular format (Norway) or graphically (Canada). However, such approaches still assume that forecasters are able to classify information about all factors unambiguously.

Our results suggest that communication of non-extreme situations is most subject to ambiguity and lack of information. Since these situations are also where most recreationalists are involved in accidents, exploring how avalanche danger is interpreted and used in decision making by users is most important here (e.g. St. Clair et al., 2021; Finn, 2020).

## 6.3 Limitations

We explored danger descriptions from avalanche forecasts published in Switzerland, using a structured sentence catalogue, where the operational language used by forecasters was German. Thus, care is required in generalising our results to forecasts published by other warning services in other languages, even where the same avalanche danger scale is in use. For instance, Techel et al. (2018) noted a different use of danger level 4-High in France and parts of Italy, which may indicate that these are interpreted in a slightly different way compared to forecasts issued in Switzerland or Austria.

Words and phrases were annotated by the forecasting team, who always work in pairs, and thus are very familiar with both the structured sentence catalogue and the avalanche situation in Switzerland. Our inter-annotator agreement is therefore likely to be higher than for avalanche warning services where forecasters work alone, or where free text is used to write forecasts. Furthermore, situations with 4-High and 5-Very High avalanche danger were much rarer (N=158; N=4, respectively) than the large number of danger descriptions for danger levels 1-Low to 3-Considerable.

The annotation was performed at the level of the entire set of phrases, not the list of phrases actually used in the forecasts. Thus, our approach does not distinguish whether a phrase not being used is simply due to it being typical for a rare situation (for instance describing danger level 5-Very High), or because forecasters are not in full agreement using this phrase as suggested in the EADS (for instance a single mountain climber representing a high additional load).

Finally, the structured sentence catalogue was not completely static over time (Hutter, 2020). Although we took account of changes to the available words and phrases, these changes (and changes to the various definitions used for, for example, avalanche size, Tab. A6; EAWS, 2019) are likely to have influenced the interpretations made by the avalanche forecasting team in their annotation, and to the use of language in forecast.

## 7 Conclusions

We analysed the text describing the expected avalanche conditions in almost 6000 danger descriptions, written relying on a catalogue of phrases, published in the public avalanche forecast in Switzerland. We focused specifically on three factors described in the textual danger description: the type of trigger required to release an avalanche, the frequency of potential triggering spots, and the expected largest avalanche size, and their relation with the issued danger level. To conduct this analysis, the Swiss avalanche forecasters assigned categories to individual terms used in the danger description. Although the

agreement in the labeling was rather high - 50% of the terms were assigned to the same class by all eight annotators, not all terms could be assigned to a specific class by some forecasters.

When we linked the factors used in danger descriptions to avalanche danger we found that, especially for 2-Moderate avalanche danger, only 21% of descriptions used all three factors and 30% of descriptions mentioned only a single factor. Furthermore, avalanche size classes are used differently to describe dry-snow and wet-snow or gliding avalanche conditions. The results highlight the demand to review and harmonize the use of terms to describe the trigger required to release an avalanche, the frequency of potential triggering spots, and the expected largest avalanche size, and their relation to the danger level. Since our approach is data driven, it provides very clear pointers as to terms which are used inconsistently or not at all by forecasters, and thus gives a valuable framework for forecasting services in reviewing the quality and consistency of written forecasts.

However, we focused exclusively on the perspective of Swiss forecasters working in German. Our results cannot be directly transferred to other forecasting services and languages, and the analysis was greatly simplified by use of the sentence catalogue used to write Swiss avalanche danger descriptions.

Future work should also explore the perspective of the user of the avalanche forecast. Are danger descriptions the best-possible way to communicate important pieces of information including avalanche size or the occurrence of natural avalanches? Do users interpret this information in similar ways to forecaster?

*Code and data availability.* Code and data will be made available at the data repository of the Swiss Federal Institute for Forest Snow and Landscape Research WSL www.envidat.org.

*Author contributions.* This contribution is based on the MSc thesis by VH (Hutter, 2020). VH developed the study design, performed the annotation of the values and the initial analysis. FT and RP supervised the MSc thesis. For the purpose of this manuscript, FT re-analysed a subset of the data used in the MSc thesis, and conducted the survey of the avalanche forecasters. FT and RP wrote the manuscript, VH provided repeated feedback on the manuscript.

*Competing interests.* None.

*Acknowledgements.* We thank Benjamin Zweifel, Célia Lucas, Christine Pielmeier, Jürg Trachsel, Kurt Winkler, Lukas Dürr and Thomas Stucki, avalanche forecasters at the national avalanche warning service at the WSL Institute for Snow and Avalanche Research SLF in Switzerland, for their annotations. We greatly appreciate the detailed feedback provided by the two anonymous reviewers. We thank Christoph Mitterer (avalanche warning service Tyrol/Austria) and Thomas Feistl (avalanche warning service Bavaria/Germany) for providing feedback

regarding the structure and content of their forecast products (Table 1). We also thank the two anonymous reviewers and the editor Pascal Haegeli for providing constructive feedback, which helped us to improve this manuscript.

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

# Appendix A: Annotation rules, German-English word lists

In the following, we first provide a short version of the annotation rules (Tab. A1).

Tables A2 to A5 list the German words and their English translation for the three key factors and their labels. These may be used as a guidance for the interpretation of the labels used in the Swiss avalanche forecasts. The proportions shown in brackets indicate how often a German word was assigned to a label. For instance, for *maintaining distances between individuals*[12] (0.75) (Tab. A2), 6 of the 8 forecasters considered this to mean a *low additional load*. Note that the proportions shown in the English column are indications only, as many-to-one, and one-to-many translations were possible (i.e. two English translations for one

German word as for *spontan* (German) and English *spontaneous* or *occur naturally*.

    Finally, Table A6 shows the labels for the avalanche size classification prior to their renaming in 2019.

**Table A1.** Annotation rules.

| key factor | Assign expressions which indicate... |
|---|---|
| type of trigger or release probability | ... the occurrence of natural avalanches (e.g. *spontaneous*) or the probability of an avalanche release (e.g. *to be expected*) or the trigger required to release an avalanche (e.g. *human*). Consider also temporal aspects (e.g. *avalanches are possible any time*). |
| frequency and location of triggering spots | ... the frequency or location of triggering spots. Distinguish between terms which indicate a frequency or number (e.g. *many* → frequency), and a location in the terrain (e.g. *close to ridge line* → location) describing triggering spots. |
| avalanche size | ... an avalanche size. Consider terms officially defined by EAWS (Tab. A6, e.g. *large*, 2019), but also those which may be considered synonyms or placeholders for an avalanche size (e.g. *fairly large*). |

---

[12]German original: *Entlastungsabstände*

**Table A2.** Labels assigned to German text values describing the type of trigger (or load) to release an avalanche, including the proportion of forecasters who assigned the respective label - sub-label combination to this value (in brackets), and their corresponding English translations. Note that the proportions shown in the English column are indications only, as many-to-one, and one-to-many translations were possible (i.e. one English translation for two German words, or vice versa). Bold - value was used during the eight years.

| key factor | label | sub-label | German | English |
|---|---|---|---|---|
| type of trigger | natural | | **spontan** (1); **jederzeit möglich** (1); **möglich** (1); **zu erwarten** (1) | **spontaneous / occur naturally** (1); **anytime possible** (1); **possible**; **to be expected, probable** (1) |
| | additional load | low | **einzelner Wintersportler** (1); **Person** (0.88); **mit kleiner Belastung** (1); **störanfällig** (0.75); **können sehr leicht ausgelöst werden** (1); **können leicht ausgelöst werden** (1); **leicht auslösbar** (1); **Entlastungsabstände** (0.75) | **single winter-sport participants** (1); **person / human** (0.88); **even in case of small load** (1); **prone to triggering** (0.75); **can be released very easily** (1); **can be released easily** (1); **maintaining distances between individuals** (0.75) |
| | | low or high | **auslösbar** (0.75); **können ausgelöst werden** (0.75); Bergsteiger (0.63); Fussgänger (0.63) | **capable of being triggered** (0.75); **can be released** (0.75); climber (0.63); hiker (0.63) |
| | | high | **mit grosser Belastung** (1); Gruppe Personen (1); Sprengung (1); gesprengt (1); kaum auslösbar (1) | **with large additional load** (1); group of people (1); explosives-triggered (1); unlikely to be released (1) |

**Table A3.** Labels assigned to German text values describing the frequency of potential triggering spots or the number of avalanches, including the proportion of forecasters who assigned the respective label to this value (in brackets), and their corresponding English translations. Note that the proportions shown in the English column are indications only, as many-to-one, and one-to-many translations were possible (i.e. one English translation for two German words, or vice versa). Bold - value was used during the eight years.

| key factor | label | German | English |
|---|---|---|---|
| frequency | low / a few | **sehr vereinzelt** (1); **vereinzelt** (1); **lokal** (0.88); **sehr selten** (1); **selten** (1); **eher selten** (0.63); **nur wenige** (1); **wenige** (1); **einzelne** (1) | **very isolated** (1); **isolated** (1); **in some localities** (0.88); **very / rather rare** (1); **rare** (1); **rather rare** (0.63); **rather few / a few** (1); **few / a few** (1); **locally** (1) |
| | medium / some | **teilweise** (1); **teils** (1); **stellenweise** (0.63); mehrere (0.88) | **in some cases** (1); **in some places** (1); several (0.88) |
| | high / many | **sehr viele** (1); **viele** (0.88); **zahlreiche** (1); **weit verbreitet** (1); **verbreitet** (0.63); **viele Stellen** (0.88); **vielerorts / an vielen Orten** (0.88); **allgemein** (0.63); **sehr / recht häufig** (1); **häufig** (0.63); **sehr oft** (1) | **a great many** (1); **many** (0.88); **numerous** (1); **very widespread** (1); **widespread / over a wide area** (0.63); **many locations** (0.88); **in many places** (0.88); **very / rather frequent** (1); **frequent** (0.63); **very often** (1) |

**Table A4.** Labels assigned to German text values describing avalanche size, including the proportion of forecasters who assigned the respective label to this value (in brackets), and their corresponding English translations. Not shown are values describing avalanche size, which could not be assigned to a label (step 1, Sect. 4.1). Note that the proportions shown in the English column are indications only, as many-to-one, and one-to-many translations were possible (i.e. one English translation for two German words, or vice versa). Bold - value was used during the eight years.

| key factor | label | German | English |
|---|---|---|---|
| avalanche size | size 1 | **kleine Lawine** (1); **Rutsch** (1); **Mitreiss- und Absturzgefahr** (1) | **small avalanche** (1); **sluff** (1); **danger of avalanches sweeping people along and giving rise to falls** (1) |
| | size 1 or 2 | **eher klein** (0.5); **nebst Verschüttungsgefahr vor allem Mitreiss- und Absturzgefahr beachten** (0.75) | **rather small** (0.5); **apart from the danger of being buried, restraint should be exercised in view of the danger of avalanches sweeping people along and giving rise to falls** (0.75) |
| | size 2 | **mittlere Lawine** (1) | **medium-sized avalanche** (1) |
| | size 2 or size 3 | **recht gross** (0.75), **ziemlich gross** (0.88), **gefährlich gross** (0.63) | **fairly large** (0.75), **rather large** (0.88), **dangerously large** (0.63) |
| | size 3 | **grosse Lawine** (1) | **large avalanche** (1) |
| | size 4 | **sehr grosse Lawine** (1); **Tallawine** (0.63); **bis in Tallagen** (0.63); grosse Tallawine (0.75) | **very large avalanche** (1); **avalanches capable of reaching the valley** (0.63); large avalanches capable of reaching the valley (0.75) |
| | size 4 or size 5 | **können ins Grüne vorstossen** (0.5) | **capable of reaching a long way into areas with no snow cover** (0.5) |
| | size 5 | **extrem grosse Lawine** (1); **können sehr/aussergewöhnlich weit vorstossen** (0.63); ausserordentlich gross (0.75); sehr grosse Tallawine (0.75); Lawinen, welche die üblichen Lawinenzüge in Länge oder Breite übertreffen (0.75) | **extremely large avalanche** (1); **avalanches capable of exceeding the length or width of the usual paths** (0.63); exceptionally large (0.75); very large avalanche capable of reaching the valley (0.75); capable of reaching a very / an exceptionally long way (0.75) |

**Table A5.** Text values providing location-specific information. Numerous combinations and variants exist. Not shown are values describing aspect and elevation, as these are normally shown in the aspect-elevation plot (cf. Fig. 1b), and values which were not used in the analysed forecasts. This list should therefore be seen as an example rather than an exhaustive list.

| German | English |
| --- | --- |
| kammfern | at a distance from ridgelines |
| windgeschützte Lagen | protected from the wind |
| Geländekanten | behind abrupt changes in the terrain |
| Felswandfüsse | base of rock walls |
| Passlagen | pass areas |
| Kammlagen | adjacent to ridgelines |
| Gipfellagen | the vicinity of peaks |
| Böschungen | cut slopes |
| Grashänge | grassy slopes |
| felsdurchsetztes / absturzgefährdetes Gelände | rocky terrain / in terrain where there is a danger of falling |
| (steile / sehr steile / extrem steile) Hänge / Gelände | (steep / very steep / extremely steep) slopes / terrain |
| Übergänge von (wenig zu viel Schnee) | at transitions from a shallow to a deep snowpack |
| in Randbereichen | at their margins |
| bei Einfahrt in (Rinnen / Mulden) | when entering (gullies / bowls) |
| schneearme Stellen | where the snow cover is rather shallow |
| Triebschneehänge | wind-loaded slopes |
| (hoch gelegenen / (noch nicht entladenen) Einzugs-gebieten | (high-altitude) starting zones (that have retained the snow thus far) |
| häufig befahrenes Variantengelände und Tourengelände | highly frequented off-piste terrain and on popular back-country touring routes |
| selten befahrenes Gelände | in little used terrain |
| Waldgrenze | at tree line |

**Table A6.** Shift in textual labels assigned to avalanche sizes according to the European avalanche Warning Services (EAWS) in 2018 (until 2018: SLF (2017), since winter 2018/19: (EAWS, 2019; SLF, 2019).

| size class | label | |
|:---:|:---:|:---:|
| | until 2018 | since winter 2018/19 |
| 1 | sluff | sluff, small |
| 2 | small | medium |
| 3 | medium | large |
| 4 | large | very large |
| 5 | very large | extremely large |