# Peer review of "How is avalanche danger described in textual descriptions in avalanche forecasts in Switzerland? Consistency between forecasters and avalanche danger"

_Natural Hazards and Earth System Sciences, 2021_

## Author Response (AR1)

*Dear editor, dear reviewers*

*Thank you for your constructive comments. We have made extensive revisions to the manuscript to deal with these, which we describe in detail below.*

*The major additions include:*

- *Table 1: This table shows examples highlighting how the information describing the three key factors are presented in different forecasts in North America and Europe.*
- *Table 5: This table summarizes our findings and is intended to facilitate comparison with the description in the European Avalanche Danger Scale*
- *Section 2: describing in detail the public avalanche forecast in Switzerland*
- *Section 6.1: implication of the findings for forecasters*

*Please find below a point by point reply to each of your comments, and how we have addressed these in the revised manuscript (blue, italics).*

**\*\*\*\*\*\*\*\*\*\*\*\*\*\*\*\*\*\*\*\*\*\*\*\*\*\*\*\*\*\*\*\*\*\*\*\*\*\*\*\*\*\*\*\*\*\*\*\*\*\*\*\*\***

**1ˢᵗ Reviewer**

**Major Comments**

*Link between theoretical background and analytic approach*

- I appreciate the introduction of the semiotic triangle as a conceptual framework for the task of avalanche forecast production. As the authors point out (lines 41-44), the semiotic triangle is helpful in that it tracks the process from an avalanche situation to forecaster interpretation to a communication. However, I am not able to see the connection from this conceptual framework to the methodological approach. To make this connection stronger, I recommend that the authors revise their introduction to present their research questions and objective in a more accurate way.

*As we believe that the semiotic triangle does have a connection with avalanche forecasting, and particularly with regard to the communication of avalanche conditions, we kept this concept. However, we now focus on consistency in the Introduction and introduce the semiotic triangle in the Discussion (Sect. 5, 299-309). When discussing the findings, we make the link to the semiotic triangle.*

*Clarifying the stated objective*

- To add more detail to the comment above, the stated objective of this study requires further clarification. If the objective of the research is to demonstrate the value of text-based analysis to avalanche forecast research (lines 54 and

66-68), the authors need better situate and justify their rationale for the study design and analysis within the body of literature on text-based methodologies. As the methods section does not include any citations to support the methodological approach beyond validating the inter-rater agreement rate (lines 169-170), the authors need to provide more adequate support to ensure that the study is well-grounded and that the reader can see how it makes a contribution to the stated objective.

*To support the choice of the methodological approach, we now briefly introduce the theoretical framework in the introduction (81-83) and provide more detail including references in the Methods section (131-137).*

- If this extends beyond the possibilities of the current analysis, I recommend that the authors reword the objective to make it clear that the goal is to contribute to an official translation of terms characterizing key factors of avalanche hazard rather than to demonstrate the value of text-based analysis in avalanche research.

*The objective of the study is to demonstrate how avalanche danger is described (and whether this is in line with definitions) by means of text-based analysis. For instance, without this analysis of the narrative danger descriptions, we would not have become aware that missing information is not distributed randomly (cf. Tab. 5, 345-350). While we can't answer why information is missing (375-378), we now point more clearly to potential implications to forecasters and forecast users (378-381, 392-398).*

*RQ1: An analysis of forecaster agreement may not represent language use*

- While important insights emerge from the analysis of RQ1, the task does not replicate the forecasting workflow and the implications to the avalanche forecasting process require re-examination. There is a crucial difference between the analytic exercise designed to examine RQ1 (lines 26-29) and the forecasting process outlined in the semiotic triangle. In contrast to the semiotic triangle, the analytic exercise does not replicate the forecasting task of moving from an avalanche situation to an interpretation and subsequent communication symbol. Rather, it orders this process in reverse, whereby the forecaster is tasked with matching a communication symbol to a corresponding key factor in an avalanche situation. Thus, the analytic approach does not examine how language is used by forecasters in the context of how forecasts are produced, which is what RQ1 might suggest given its current wording (i.e., "how do forecasters use language….") (lines 75-76). A more precise wording of RQ1 might read, "how well do forecasters agree on the meaning of key phrases...."

We agree regarding the formulation of RQ1, we reworded to "How well do forecasters agree on the meaning of terms characterizing triggers required to release

avalanches, frequencies of triggering spots, and expected avalanche sizes?" (73-74, and throughout the manuscript). We also moved the semiotic triangle to the discussion, and use it as a tool to explore the implications of the results.

*RQ2: Establishing a hypothesis*

- Research question 2 (lines 133-135) involves analyzing how the classified text descriptions correlate across differences in avalanche danger. The authors distinguish avalanche danger according to the different levels of severity as classified by the European Avalanche Danger Scale and according to dry-snow versus wet-snow conditions. The analysis examines a measure of the completeness of trigger, likelihood, and size information across differences in avalanche danger as well as examines their content distinguished by natural and additional load triggers; few, several, or any triggering locations; and sizes 1, 2, 3, 4, and 5.
- To establish a starting point for expected outcomes, the authors need to provide a reference to the full European Avalanche Danger Scale in the main body of the manuscript. Based on the formal definitions of the various levels, what differences in completement and content, if any, would be reasonable to expect? Providing this background and hypotheses would better situate the results in terms of how they confirm or contrast existing expectations. This would help to expand the discussion of the value of the danger description and the potential reasons for the observed variabilities.

*We now provide the full European Avalanche Danger Scale (EADS, EAWS, 2018) in Table 2. To facilitate the comparison between the EADS and our findings, we now provide Table 5 which summarizes the terms used in the EADS and the terms used to describe avalanche danger in the Swiss danger description. We now dedicate the new Section 2 explaining in more detail the avalanche forecasts in Switzerland.*

- Similarly, the authors need to include their rationale for including and differentiating dry-snow versus wet-snow avalanche types. Why do they make this distinction? Are any avalanche conditions excluded from this distinction? And finally, how is information completeness and content expected to differ across these conditions? Providing this background and hypotheses would better situate the results in terms of how they confirm or contrast existing expectations.

*We now provide more details regarding the communication of dry-snow vs. wet-snow conditions in the Swiss forecast (Sect. 2, 95-101), highlight differences by summarizing key findings in Table 5, and discussing these differences (353-368).*

*Communicating Uncertainty*

- As the authors detail in their explanation of the semiotic triangle (line 40), a key aspect of the cognitive task in avalanche forecasting is that forecasters may work through the semiotic triangle with incomplete information, which produces various levels and sources of uncertainty. Is it possible that situations of extreme danger might have different levels or sources of uncertainty than situations of moderate danger and might explain some of the resulting patterns in the analysis?

*We now briefly discuss potential explanations for missing information (375-381).*

- The authors do highlight this possibility in the discussion section (lines 338-339). However, as the element of incomplete information was pre-defined in the semiotic triangle, this very limited mention of it in the discussion seems underdeveloped and incomplete. The manuscript would benefit from elaborating on the role of incomplete information as it currently leaves a lot of questions open.

*We now elaborate on the role of incomplete information with regard to its meaning and implication to forecasters and users (345-350, 375-381, 388-398).*

RQ2: The inclusion and exclusion of phrases

- Through the analysis of RQ1, text phrases that did not produce high levels of agreement among forecasters regarding key factors were subsequently excluded from additional analyses. This begs the question: what themes were encompassed by these ambiguous phrases?

*This is an important point, and we now make clearer the iterative annotation process we used, typical of such text analysis (134-137, Sect. 4.1-4.2). For example, we now illustrate phrases which were thought to represent one of the key factors describing avalanche hazard in initial annotation, but were not assigned to an absolute class during the second annotation step (187-191).*

- Could these phrases also offer valuable insight into avalanche forecast quality? I recommend that the authors consider conducting further analysis of the excluded phrases. The results could then be incorporated into the analysis of RQ2 for a more robust analysis. Is it possible that the themes encompassed by the ambiguous phrases might correlate with specific hazard conditions and might provide insights into what forecasters deem important to danger descriptions beyond key phrases?

*Since we discarded terms where annotators could not assign a label, we cannot go beyond giving examples (187-191), since by definition these phrases were then not labelled.*

- Additionally, does it make sense to include phrases in the analysis that were never used in a bulletin? This should be addressed in the limitations section.

*The annotation was performed at the level of the entire set of phrases, not the list of phrases used in the forecasts. We added a statement in that regard in the limitations section (413-416). However, it is important to note that our characterisation (RQ2) is based on phrases used in bulletins.*

*Implications for users of avalanche forecasts*

- Line 338: "Leaving out information, for example the likely triggers or size classes of avalanches expected for danger level 2-Moderate, may, for forecasters, actually convey information about the situation." Please elaborate on this. Maybe provide an example.

*We now provide an example(390-395).*

- There are various papers, such as Lazar et al. (2016), Statham et al. (2018), and Clark (2018 and 2019) that shed light on consistencies or inconsistencies among avalanche forecasters. I think it would be useful for this paper to include these ISSW papers in the discussion.

*Thank you for pointing these out. We incorporated these when discussing recent work on consistency in avalanche forecasts (43-45).*

- The discussion does not include any recommendation for avalanche forecasters or the Swiss avalanche bulletin system (e.g., use of phrase catalogue). While there is a brief mentioning of the graphic display of avalanche hazard information in Canadian avalanche bulletins, a critical discussion of how the graphical approach and/or the conceptual model of avalanche hazard (Statham et al., 2018) can address the identified challenges is missing. I believe that a broader discussion would make this a more useful paper for the global avalanche safety community.

*In the introduction, we now provide a new table (Table 1) providing an overview showing which forecast components are displayed, and how. We now refer to this overview at several locations in the manuscript (32-38, 395-398). Furthermore, we added a Section 6.1 (Implications to forecasters) in the Discussion, where we make recommendations for Swiss forecasters, and for forecasters with the same working language. We now also make a link to the CMAH, pointing out potential benefits of following this approach, but also issues which must be addressed (378-381).*

*Limitations*

- Given that the use of the sentence catalogue seems to be very specific to the production of the Swiss avalanche bulletin, I don't think it is realistic to expect

that the results would be transferable to other warning services. I believe that the focus on Switzerland should be clearly stated in the research objectives. This means that this aspect likely does not need to be mentioned in the limitations section.

*The results are clearly specific to the Swiss forecast even though the sentence catalogue is used by five warning services in Europe. We emphasize that we focus on Switzerland in the research objective (75). Furthermore, we now dedicate Section 2 introducing the public avalanche forecast in Switzerland. But we still believe that not being able to transfer the findings easily to other forecast products is a limitation (403-408).*

- Per my earlier comment on the inclusion and exclusion of phrases in the analysis, I believe that this should be addressed in the limitation section.

*We discuss this now on 413-416.*

**Minor Comments**

*Triggering terminology*

- I find the terms used to describe the key factors related to triggering avalanches to be wordy and confusing (i.e. triggering leve, triggering spots frequency, and triggering spots location). I think the following terms from the Conceptual Model of Avalanche Hazard (Statham et al., 2018) offer a clearer delineation of these key elements: trigger type, sensitivity, and spatial distribution. These elements are then combined to form the likelihood of avalanches, whereas the size classifications offer an ordinal representation of consequence. I recommend the use of these terms as it strengthens the connection to well-established definitions of key factors within risk science.

*As the basis for forecasting in Europe and Switzerland is primarily the EADS, we will stick with well-established terms currently in use in Europe (179-183, Tab. 4). However, we now make a stronger reference to terms used in the CMAH (as in Tab. 4).*

*Introduction*

- The introduction is fully focused on European avalanche bulletins. Since the authors refer to non-European avalanche bulletin formats in the discussion section, I think the manuscript would benefit from including a more in-depth description of how the information presentation in the Swiss bulletin compares to others. For example, the text information (avalanche activity, snowpack conditions, weather) included in Canadian and US bulletin offers more detailed insight about conditions than the text included in Swiss bulletins.

Furthermore, a broader description of the context in the introduction will make the paper more relevant for a wider audience.

*Using examples (Table 1 and supplementary material), we now provide an overview of how the three contributing factors of avalanche hazard are addressed in forecast products issued by different warning services in North America and Europe (i.e. graphics, bullet list, danger description), and how the three contributing factors of avalanche hazard are referred to. This will highlight more clearly differences between the Swiss forecast and other forecasts (32-38, 63, 106-107).*

- Line 63: Clark (2019) examined the link between the likelihood and expected size of avalanches with the avalanche danger rating. The manuscript does not accurately describe this research.

*We have revised accordingly (56-59).*

- Line 106: At the core of the danger description "in Switzerland"…

*This sentence does not exist anymore.*

*Discussion*

- Line 346: I do not understand how the results of the analysis suggest that "communication of non-extreme situations is critical". This statement requires elaboration.

*We rephrased and elaborate (399-401).*

**Technical Comments**

- Abstract is quite long.

  *We shortened somewhat.*

- Replace "firstly" with "first", and "secondly" with "second" and so on (e.g., Line 8, but many others as well).

  *Done throughout the manuscript.*

- Line 163: Extra ")" that is not necessary.

  *Done.*

- Line 166: "More than 20 of the values …". Please be precise?

  *Changed to 22.*

- Line 164-171: In both cases, 53% of the groups were assigned the same by all participants. Is this correct or a typo?

  *We checked this again, and these values are by coincidence identical.*

- Figure 4: Given that the lowest value on this chart is 0.64, a different color scale would bring out differences more clearly. Given these details, can the authors explain the observed differences between the participating forecasters?

  *We changed the colour scale in Fig. 4.*

- Line 187-192: No need to repeat information that is already presented in Table 2.

  *We removed these lines.*

- Line 195: Replace "All analysis was …" with either "All analyses were …" or "The entire analysis was …"

  *Done*

- Line 201: "In the descriptionS …" (missing s)

  *Done*

- Lines 201-206: The simultaneous description of the results and the example shown in Figure 5 makes the text quite convoluted. I recommend separating the two aspects to make the text more readable. Furthermore, I think that the description of the example should actually be included in the methods section, where there is already a reference to Figure 5 on Line 180.

  *We moved these examples to the section where we explain the analysis of the danger descriptions (204-210).*

- Line 207-208: The current statement does not state that the proportion of descriptions that include all three factors decreases with "decreasing" danger levels.

  *Thank you for pointing this out. We revised accordingly (234-235).*

- Line 207-211: It seems to me that this description actually belongs to the next paragraph as it already discusses the danger description at different danger levels.

  *We now only use one sub-section to describe dry-snow avalanche conditions (227).*

- Table 3: Tables cannot have shading. This makes them figures. Also note that some of the lines have been erased by the shading.

  *Now presented as Figure 6.*

- Table 3: It would be best to use a consistent format for presenting the results. The authors currently use percentages in the text while using proportions in the tables and figures.

  *Good point. We changed to proportions throughout the manuscript.*

- Figure 6: Legends should not be plotted over top of stacked bars. In addition, labeling the individual charts with titles would make the figure easier to read.

  *The legend positions have been changed and titles have been added to Fig. 7.*

- Line 278: Should be "classified" instead of "classed".

  *Done*

- Line 283: Should be Zooming "into" instead of "in to".

  *Done*

- Line 333: Why reasonably in brackets? It would be best if the authors quantified what they mean by "reasonable."

  *Removed as we rewrote this part of the discussion section.*

- Line 341: Avalanche warning services in Canada, the United States, and New Zealand are using graphical representations of the critical information.

  *We rephrased. In addition, we now provide Table 1 to highlight variations between forecast products.*

- Table A3 indicates that not all phrases have been used during the study period. This is an important detail that is not mentioned in the text.

  *This is correct. We now mention this fact (329-330, 413-416).*

**References**

Clark, T., & Haegeli, P. (2018). Establishing the link between the Conceptual Model of Avalanche Hazard and the North American Public Avalanche Danger Scale: Initial Explorations from Canada. *Proceedings of the 2018 International Snow Science Workshop*, Innsbruck, Austria, 1116-1120. https://arc.lib.montana.edu/snow-science/item.php?id=2718

Clark, T. (2019). *Exploring the link between the Conceptual Model of Avalanche Hazard and the North American Public Avalanche Danger Scale*. M.R.M. research project, no.604, Burnaby, BC. School of Resource and Environmental Management, Simon Fraser University. https://summit.sfu.ca/item/18786

EAWS: European Avalanche Danger Scale (2018/19), https://www.avalanches.org/wp-content/uploads/2019/05/European_Avalanche_Danger_Scale-EAWS.pdf, last access: 14 Feb 2020, 2018.

Lazar, B., Trautman, S., Cooperstein, M., Greene, E., & Birkland, K. (2016). North American Danger Scale: Are Public Backcountry Forecasters Applying It Consistently? *Proceedings of the 2016 International Snow Science Workshop*. Breckenridge, CO, 457-465. https://arc.lib.montana.edu/snow-science/item.php?id=2307

Statham, G., Haegeli, P., Greene, E., Birkeland, K., Israelson, C., Tremper, B., & Kelly, J. (2018). A conceptual model of avalanche hazard. *Natural Hazards*, 90(2), 663-691. https://doi.org/10.1007/s11069-017-3070-5

Statham, G., Holeczi, S., & Shandro, B. (2018). Consistency and accuracy of public avalanche forecasts in Western Canada. *Proceedings of the 2018 International Snow Science Workshop*, Innsbruck, Austria, 1491-1495. https://arc.lib.montana.edu/snow-science/item.php?id=2806

**\*\*\*\*\*\*\*\*\*\*\*\*\*\*\*\*\*\*\*\*\*\*\*\*\*\*\*\*\*\*\*\*\*\*\*\*\*\*\*\*\*\*\*\*\*\*\*\*\*\*\*\*\*\*\*\*\***

**2nd Reviewer**

The manuscript reads little like it was extracted from a larger document. There is both too much detail and use of very specific terms (like phrase_option) in some sections and not enough background and explanation in others (see note on semiotic triangle below). This issue runs through many parts of the manuscript. It makes interpreting some of the figures difficult as they are dense with information, but don't provide the reader with much guidance on what is included to provide a holistic view of the work and what is critical to understanding it (Table 3 is a good example).

*We now provide more guidance on how to read figures and tables. For instance, we provide more detail in the caption to Figure 6 (which was Table 3 before), and refer to specific lines in this figure in the text (235, 238, ...)*

The Methods section includes descriptions of which author produced different parts of the analysis, although I'm not sure if this is important for me to know and if so why.

*We removed the author initials in the Methods section.*

The Data section explains a lot about how the avalanche forecasts are issued and identifies elements of the products, but it does not paint a clear picture of how the structure of the phrase catalogue informs or affects the analysis or results.

*The catalogue of phrases impacts the forecast product as all forecasters use the same set of words. It also impacts the analysis as the number of words, though quite large, is finite. Only this combination makes the research possible. We added these details to the manuscript (78-80).*

The Results section is very dense with a lot of information, but sorting through it is challenging (thank you for the summaries by danger level!).

*We rephrased the Results section (Sect. 5, 225ff) with the intention to make it easier readable (although it is still dense). In addition, we now provide a table (Table 5) summarizing the key results together with the description used in the European Avalanche Danger Scale (EAWS, 2018) for better comparison between our findings and the definitions in the danger scale.*

I am both intrigued and confused by the appendix, which has lots of information that is interesting but I'm not sure how important it is for understanding or documenting the work.

*We consider the appendix as complementary information: for researchers who intend to reproduce such work, but also for those who understand both German and English. Importantly, the analysis was fully conducted on German text, and we consider it important to make clear the (possible) implications of translation, without impacting on the readability of the main body of the paper. We now provide more explanation with the tables in the Appendix to facilitate their understanding (546-554), but also in the respective table captions A2-A5.*

The text in the manuscript could be improved. It is free of errors and typos. However, the writing does not always help the reader focus on the most important issues faced by the researchers or highlighted by the research. An example is paragraph 40. It is rich with ideas and constructed so it is hard for the reader to mover trough it smoothly.

*We have removed the paragraph on the semiotic triangle in the Introduction (lines 40 ff). We made an effort to focus more clearly on the key aspects of the study throughout the manuscript.*

Here are a few specific comments:

Title – The subtitle feel closer a description of what is contained in the manuscript. The main title is quite broad and I do not feel like it really helps the reader know what to expect within the article. I suggest they authors refine the title and subtitle structure.

*We rephrased the title to "How is avalanche danger described in textual descriptions in avalanche forecasts in Switzerland? Consistency between forecasters and avalanche danger".*

Terminology – The authors use some terms in ways that are consistent with previous work, but also that are probably not part of current general use for most of the readers they are trying to reach.

*We introduce the terms more clearly, including their origin (179-184). We stick to terms used in European documents (i.e. the European Avalanche Danger Scale) as these are the binding guidelines in European avalanche forecasting (i.e. 178-182). However, we link these terms to the Conceptual Model of Avalanche Hazard (CMAH; Statham et al., 2018), which is increasingly used in Europe as well (Table 4 showing the key factors and their labels together with the respective key factor term used in the CMAH; also in the new Table 1 where we are summarizing forecast examples).*

They do dedicate a paragraph in the introduction to explain the linguistic model they are applying, which is admirable. However, I found the layout and use of some terms in the manuscript confusing. One example is symbol. This is important to the model the author's use and it is also commonly used in warning communication. The authors use it in the context of the linguistics model and also to refer to graphical elements. The paragraph on the linguistics model contains a lot of good information, but the liberal use of parenthetical phrases makes the material difficult to digest. My suggestion is to do one of two things: remove the discussion of the semiotic triangle and associated ideas and focus on the consistency issues in the stated research questions, or expand the discussion of the semiotic triangle and associated concepts. If this concept is integral to the work, maybe it deserves its own section with a clear explanation. Applying this concept to avalanche forecasts is certainly interesting, but I am not sure if it is fundamental to understanding the work. To me the work described in this manuscript focuses on issues of forecast consistency (consistency of elements within a forecast). If the authors opt to keep the concepts of the semiotic triangle, I suggest they take some time in the proposed section to clearly define how the terms and concepts in this linguistic model are represented in the avalanche forecasts they are analyzing.

*We now focus on consistency in the use of the terms, between forecasters and when compared to the EADS. We now introduce and apply the semiotic triangle in the Discussion section only (Sect. 6, 299ff). However, we consider it a highly relevant concept for the interpretation of our findings, and refer to it throughout the discussion section (312-316, 342-344, 383-384).*

Focus of the results – This work is very specific to the public avalanche forecasts in Switzerland. The authors acknowledge this in the title. In many other parts of the manuscript, the text in not as specific and often is phrased in a way that makes the reader feel like they are learning about avalanche forecasts in a broad sense. This could, and should be improved. The authors should focus on the Swiss products. This study would probably not be possible with a broader dataset. This provides the authors opertunity to focus on specific aspects of the dataset and intrerpret the results in a realistic and targeted fashion.

*We are now more explicit that we are analyzing Swiss forecasts. We dedicate a new Section 2 on the avalanche forecast in Switzerland (85ff). In addition, we provide examples for other forecast products to highlight where the forecast in Switzerland differs compared to other forecasts (Table 1).*

Last sentence of abstract – "Our results provide data-driven insights that could be used to refine the ways in which avalanche danger could and should be communicated, especially to recreationalists, and provide a starting point for future studies on how users interpret avalanche forecasts." These are very important issues and certainly worth studying and improving. However, I don't see how that is done in this work. The work focuses on the internal consistency within an avalanche forecast – text descriptor and avalanche danger. It really doesn't tackle how avalanche danger or the threat to a person could or should be communicated. Just the consistency within the public product in Switzerland. This is a study of how avalanche danger IS being communicated. Given that internal consistency is an important element of any warning product, this work could be a measure of the effectiveness of that product from the producer's perspective (ie consistent elements are important and reduce the potential of confusing of the target audience). However, there is no measure of how the target user is accepting, comprehending, or effectively applying the warning product.

*We have removed this part from the abstract. We discuss the potential impact of missing information for forecast users in Section 6.2.*

**References:**

EAWS: European Avalanche Danger Scale (2018/19), https://www.avalanches.org/wp-content/uploads/2019/05/European_Avalanche_Danger_Scale-EAWS.pdf, last access: 14 Feb 2020, 2018.

Statham, G., Haegeli, P., Greene, E., Birkeland, K., Israelson, C., Tremper, B., & Kelly, J. (2018). A conceptual model of avalanche hazard. *Natural Hazards*, 90(2), 663-691. https://doi.org/10.1007/s11069-017-3070-5

\*\*\*\*\*\*\*\*\*\*\*\*\*\*\*\*\*\*\*\*\*\*\*\*\*\*\*\*\*\*\*\*\*\*\*\*\*\*\*\*\*\*\*\*\*\*\*\*\*\*\*\*\*\*\*\*\*\*\*\*\*\*\*\*

**Editor**

Dear authors:

In addition to what the reviewers have provided, I have a few additional comments/suggestions that have not come up yet:

RQ1: To me, RQ1 relates much more to how SLF forecaster perceive the existing terms included in the phrase catalog and the consistency of this perceptions and less about the use of these terms. I think that this wording is much more consistent with what you describe in the methods section and the content of the results and discussion sections.

*We agree. We have rephrased RQ1 to " How well do forecasters agree on the meaning ofterms characterizing triggers required to release avalanches, frequencies of triggering spots, and expected avalanche sizes?".*

Lines 138-152: Author initials in method section do not seem necessary. Authors contributions are better described in the author contribution section than in the methods section.

*We removed the author initials.*

Line 181: What are the implications of the randomly selected labels on the results? Would always opting for the more unfavourable level have changed the results? This could be discussed in the results or limitation section.

*We now describe this influence in the results section (268-275).*

Line 193-196: In the results section, you seem to describe several trend analyses that compare more than two groups or cover the entire danger rating level range (e.g., Line 217: "The proportion of danger descriptions which mentioned natural avalanche occurrence increased significantly from one danger level to the next higher (1-Low to 4-High, p < 0.001)."). How are these analyzed? This is not described in the methods section.

*We always calculated the p-value for pairs, not for a sequence. We hope that we now explain this short-cut of stating one p-value (rather than three in this case) in the Methods section (219-224).*

Line 193-196: Since you are doing many pair-wise comparisons, it seems that your p-values should be corrected (e.g., Bonferroni correction) to counteract the problem of multiple comparisons. I believe that the pairwise.prop.test() function in R does this automatically. However, since you are only interested in sequential comparisons

(e.g., Low vs Mod, Mode vs Cons) it is better probably to calculate the uncorrected p-values first and then adjust them with the p.adjust() function.

*We calculated pairwise p-values for sequential danger levels by looping through danger level pairs. We describe this now more clearly on 219-224.*

Line 218-221: In the discussion of the probability of natural release terms, you are suddenly examining the original terms again. This is a little bit unexpected because you do not seem to do this in other areas. Furthermore, it might be useful to indicate that the expected and probable terms are the same in German (At least, this is what I read out of Table A4).

*We now introduce these terms (183-184, Table 4) and specifically show that expected and probable has only one translation in the catalogue of phrases (247-248). In addition, we mention in the explanations to the tables in the Appendix that there are many-to-one and one-to-many translations between German and English.*

Line 240-244: How you came up with this summary has not been described anywhere.

*We describe this now in the captions of Table 5.*

Tables A3 and A4: Given that the majority of the NHESS readership is not German speaking, I think it would be useful to include proportions in the English column of the tables as well. Non-German speakers might not be able to connect the German and English terms very easily. In addition, the terms 'German' and 'English' should always be capitalized.

*We now provide proportions in the English column as well (Tables A2 - A4). however, we had to make a note in the table captions that the proportions in the English column should be considered an indication only, as one-to-many or many-to-one translations are possible.*

Table A3 and A4: Based on your explanation on Line 172 ("If five annotators (a majority) indicated the same class, the value was assigned to this class. If there was no clear majority vote, the value was assigned to the two most frequent classes chosen."), shouldn't terms with proportions of less than 0.63 (5 of 8) appear twice in these tables. As far as I can tell, "Tallawine (0.38)" only appears once.

*This was an error that has been corrected (Table 4, Tallawine (0.63)).*

Table A4: In the frequency section, many terms are associated with ranges of forecaster proportions. While I assume that this is related to the terms in brackets, it is unclear to me how they are connected. Explicitly writing out each term might be clearer and easier for the reader to understand.

*We now explicitly write out each term, including the respective proportions. Note that one-to-many or many-to-one translations are possible.*

---

## Author Response (AR2)

*Dear Pascal*

*Thank you for again providing feedback. Please find below a point-by-point response.*

*Best regards*

*Frank*

**Non-public comments to the author**:

Dear Frank, Ross, and Veronica:

Thank you very much for the submission of your revised manuscript and for taking the time to address all of the reviewers' comments. I believe that the clarity and focus of the manuscript has improved substantially.

Please see below for some technical comments about the writing style and potential typos. I hope these comments are useful and help you finalize the manuscript. I am also highlighting some of the NHESS style guidelines you might want to consider before submitting the final production files. Taking care of these issues ahead of time will make for a smoother process.

*We now place the footnotes with the German original in brackets within the text. In addition, we now only provide translations for cases, which are not listed in the tables in the Appendix.*

*We removed the vertical lines in most of the tables.*

*Table 5: following correspondence with the editorial team when submitting the revised version of the manuscript, we refer to this table as table rather than figure. However, we provide the table as a PDF figure. While we consider that keeping the colored cells is important to facilitate the understanding of the table (similar to a heat map), we don't mind referring to this as either a table or a figure.*

In addition to these minor points, I would like to ask you to carefully consider whether all tables and figures are necessary for conveying the key take home messages of your study. The number of tables and figures seems quite large, and the content does not always seem critical and sometimes a bit redundant to what is mentioned in the text. A good example is Figure 4, which shows the kappa values between all the different forecasters. Since the agreement is substantial among all forecasters—which is the main take home message—a simple sentence stating that seems sufficient.

*We moved this table to the Appendix.*

Another example is the descriptions of Table 5 and Figure 6, where many of the statistics shown in the table and figure are explicitly mentioned in the text again. In addition to this redundancy, the frequent citing of figures and tables in the text makes the text choppy and unnecessarily difficult to follow.

*We strongly reduced the citing of figures and tables in the Result and Discussion section. We now only refer to the respective subfigures and table once in the respective Result sections 5.1 and 5.2.*

I therefore believe that a critical review of the tables, figures, and their citation in the text could improve the clarity and focus of the manuscript. However, this is more of a suggestion, and I do not have a strong opinion about this.

Minor questions

Line 46 & 51: You are using the term "symbol" here, but it is not completely clear what you mean with this. Might this be a leftover from when you explained the semiotic triangle at the beginning of the introduction? I see that you explain the term on Line 302 as words or icons. Maybe just use those terms in the introduction to make the points easier to understand with less jargon.

*We have changed these mentions to icons.*

Line 80-84: This description of the methods might not be necessary in the introduction.

*This was originally suggested by a reviewer, and we think it is helpful and would prefer to leave as is.*

Line 109: It might be easier to just say 5pm (or 17:00) local time. The time zone does not seem to matter in this context, and it gets you around the footnote.

*We have changed as requested.*

Line 222: I am not sure I understand what that sentence means.

*We have slightly reworded this sentence to make it clearer.*

Table 5: I do not think you can include shading in a table unless you turn it into a figure.

*Refer to reply by editorial team - Fig or Table.*

Table 5: I found the use of the asterisk rather confusing, and the explanation did not help me to understand what is going on. Do you really need it, or couldn't the counting of the factors be described in the text more easily?

*We don't think that the counting of the factors could be well described in detail in the text. Table 5 provides additional information. However, we changed the order of the columns 2 to 5, which allowed us to remove the asterisk.*

Line 205-208: Can you add a sentence explaining why this was useful/necessary.

*We added a clause to make this clearer :"... so as to retain the ambiguity expressed by forecasters in the analysis."*

Discussion: It seems to me that in the discussion section, you describe the proportions of term use in percentages while you express some of the same information as decimal fractions in the result section. See, for example, Line 355-357. Please check for consistency.

*We changed proportions to percentages throughout the manuscript (text, Figure 6, Table 5).*

Supplement: I am not sure what the purpose is of the supplementary material since it is not cited in your manuscript at all. Please delete if it is not critical for your manuscript.

*The supplement was intended to support Table 1, but the reference to the supplement was missing. We now refer to the supplement in the Table caption.*

Typos and minor errors

   Line 37 (and other spots): "key words" should be one word.

*Done.*

•    Line 38: delete "e.g., in Norway:" The sentence already mentions Norway.

*Done*

•    Line 51: Revise to "Engeset et al. (2018) tested the comprehension of text, symbols and picture among Norwegian avalanche bulletin users and noted …".

*Done*

•    Line 151: I don't think you need the "c.f." in this reference. In fact, I don't think this is needed in any of the references you included it.

*"c.f." is removed throughout the manuscript.*

- Line 155: Simplify to "… is provided in Appendix A1."

*Done*

- Line 203: Change "Tab.s" to "Tables".

*Done*

- Line 214: I would write "We conducted this part of the analysis in two steps: …" to better highlight that this particular description does not relate to the entire analysis.

*Done*

- Table 5: The proportion for 1 of 3* and 2-Mod is missing a decimal digit. It should be 0.30 for consistency.

*Done*

- Line 297: Simplify sentence to "To discuss our results, we introduce the semiotic triangle … "

*Done*

- Line 313: I think this should be "use" instead of "used" since this statement relates to producing avalanche forecasts in general and not just your study.

*Done*

- Line 324: I think this should be "hedge" and not "hedged".

*Done*

- Line 332: Change start of sentence to "The answer to this question provides …" because "it" could refer to other terms in the previous sentence.

*Done*

- Line 343: The "the" is not necessary and can be deleted in "… made by the forecasters …"

*Done*

- Line 407/8: You can simplify the sentence to "… is therefore likely higher …"

*Done*

- Line 435: It is unusual to start a paragraph with "however" as it is a linking phrase that ties the statement to the previous sentence.

*We have added this to the previous paragraph, as we would prefer to keep the however to clearly signal a limitation.*